# Cardiorespiratory response to early rehabilitation in critically ill adults: A secondary analysis of a randomised controlled trial

Sabrina Eggmann[1,2]*, Irina Irincheeva[3], Gere Luder[1], Martin L. Verra[1], André Moser[3], Caroline H. G. Bastiaenen[2], Stephan M. Jakob[4]

1 Department of Physiotherapy, Inselspital, Bern University Hospital, Bern, Switzerland, 2 Department of Epidemiology, Research Line Functioning, Participation and Rehabilitation CAPHRI, Maastricht University, Maastricht, The Netherlands, 3 CTU Bern, University of Bern, Bern, Switzerland, 4 Department of Intensive Care Medicine, Inselspital, Bern University Hospital, University of Bern, Bern, Switzerland

* sabrina.eggmann@insel.ch

## Abstract

### Introduction

Early rehabilitation is indicated in critically ill adults to counter functional complications. However, the physiological response to rehabilitation is poorly understood. This study aimed to determine the cardiorespiratory response to rehabilitation and to investigate the effect of explanatory variables on physiological changes during rehabilitation and recovery.

### Methods

In a prospectively planned, secondary analysis of a randomised controlled trial conducted in a tertiary, mixed intensive care unit (ICU), we analysed the 716 physiotherapy-led, pragmatic rehabilitation sessions (including exercise, cycling and mobilisation). Participants were previously functionally independent, mechanically ventilated, critically ill adults (n = 108). Physiological data (2-minute medians) were collected with standard ICU monitoring and indirect calorimetry, and their medians calculated for baseline (30min before), training (during physiotherapy) and recovery (15min after). We visualised physiological trajectories and investigated explanatory variables on their estimated effect with mixed-effects models.

### Results

This study found a large range of variation within and across participants' sessions with clinically relevant variations (>10%) occurring in more than 1 out of 4 sessions in mean arterial pressure, minute ventilation (MV) and oxygen consumption (VO$_2$), although early rehabilitation did not generally affect physiological values from baseline to training or recovery. Active patient participation increased MV (mean difference 0.7l/min [0.4–1.0, p<0.001]) and VO$_2$ (23ml/min [95%CI: 13–34, p<0.001]) during training when compared to passive participation. Similarly, session type 'mobilisation' increased heart rate (6.6bpm [2.1–11.2, p =

**Data Availability Statement:** The relevant dataset and analysis code are within the paper and its Supporting information files (S4, S5 and S6 Files).

**Funding:** This research has been funded from the PhD Grant 2018 given by the Swiss Foundation for Physiotherapy Science (http://www.physiotherapie-wissenschaften.ch/) and received by SE. The foundation had no role in study design, data collection and analysis, decision to publish, or preparation of the manuscript.

**Competing interests:** We have read the journal's policy and the authors of this manuscript have the following competing interests: SE, GL, MLV, CHGB declare that they have no competing interests. II and AM are affiliated with CTU Bern, University of Bern, which has a staff policy of not accepting honoraria or consultancy fees. However, CTU Bern is involved in design, conduct, or analysis of clinical studies funded by not-for-profit and for-profit organizations. In particular, pharmaceutical and medical device companies provide direct funding to some of these studies. For an up-to-date list of CTU Bern's conflicts of interest see http://www.ctu.unibe.ch/research/declaration_of_interest/index_eng.html. SMJ reports the following potential conflicts of interest: The Department of Intensive Care Medicine has, or has had in the past, research contracts with Orion Corporation, Abbott Nutrition International, B. Braun Medical AG, CSEM SA, Edwards Lifesciences Services GmbH, Kenta Biotech Ltd, Maquet Critical Care AB, Omnicare Clinical Research AG and research & development/ consulting contracts with Edwards Lifesciences SA, Maquet Critical Care AB, and Nestlé. The money was paid into a departmental fund; SMJ received no personal financial gain. The Department of Intensive Care Medicine has received unrestricted educational grants from the following organizations for organizing a quarterly postgraduate educational symposium, the Berner Forum for Intensive Care (until 2015): Fresenius Kabi, gsk, MSD, Lilly, Baxter, astellas, AstraZeneca, B | Braun, CSL Behring, Maquet, Novartis, Covidien, Nycomed, Pierre Fabre Pharma AG (formerly known as RobaPharm), Pfizer, Orion Pharma, Bard Medica S.A., Abbott AG, Anandic Medical Systems. The Department of Intensive Care Medicine has received unrestricted educational grants from the following organizations for organizing bi-annual postgraduate courses in the fields of critical care ultrasound, management of ECMO and mechanical ventilation: Pierre Fabre Pharma AG (formerly known as RobaPharm), Pfizer AG, Bard Medica S.A., Abbott AG, Anandic Medical Systems, PanGas AG Healthcare, Orion Pharma, Bracco, Edwards Lifesciences AG, Hamilton Medical AG, Fresenius Kabi (Schweiz) AG, Getinge Group Maquet AG, Dräger Schweiz AG, Teleflex Medical GmbH. None of these companies had a role in this study. Our relationship

0.006]) during recovery when compared to 'exercise'. Other modifiable explanatory variables included session duration, mobilisation level and daily medication, while non-modifiable variables were age, gender, body mass index and the daily Sequential Organ Failure Assessment.

## Conclusions

A large range of variation during rehabilitation and recovery mirrors the heterogenous interventions and patient reactions. This warrants close monitoring and individual tailoring, whereby the best option to stimulate a cardiorespiratory response seems to be active patient participation, shorter session durations and mobilisation.

## Trial registration

German Clinical Trials Register (DRKS) identification number: DRKS00004347, registered on 10 September 2012.

## Introduction

Intensive care unit (ICU) survivors frequently experience poor recovery with multifactorial physical, cognitive and mental health impairments [1–5]. Early rehabilitation and mobilisation in critically ill patients are advocated to attenuate these complications [6]. Their safety has been confirmed in a large meta-analysis of >22,000 rehabilitation sessions [7]. Moreover, they increase functional mobility and muscle strength [8] and reduce days with delirium and mechanical ventilation [9]. However, insufficient intervention reporting of frequency, intensity, type and timing limit their implementation in clinical practice [10]. Additionally, the appropriate exercise type or intensity to induce appropriate physiological adaptations remains yet to be determined.

There is previous evidence that sitting on the edge of the bed may be associated with higher metabolic cost compared with a passive chair transfer [11] or passive in-bed cycling [12]. The hierarchy of these mobility intensities are mirrored in the ICU mobility scale [13] although the highest activity levels may not be the most physically intensive [14, 15]. Similarly, the cardiorespiratory response is likely to vary between different types of exercise, exercise-modalities or individuals [16, 17]. Individually tailored rehabilitation interventions therefore seem appropriate. However, there is little data to guide the physiotherapist's clinical decision-making in delivering optimal training intensity.

The purpose of this planned secondary analysis was to determine the cardiorespiratory response to early physiotherapy-led rehabilitation in mechanically ventilated, critically ill adults in a mixed ICU. First, we aimed to analyse physiological variables from before, during and after rehabilitation to describe both the response to rehabilitation and to estimate recovery. Second, we aimed to characterise effects of a-priori selected explanatory variables on cardiorespiratory reactions from before to during rehabilitation and from before to after rehabilitation. We hypothesised that session type, mobilisation level, exercise modality and duration as well as individual patient characteristics would be the main drivers of the physiological response to rehabilitation. Finally, we re-evaluated safety by examining sessions with clinically relevant variations (>10%), an adverse event or therapy discontinuation to determine predictors and explored individual characteristics of patients with a strong physiological reaction.

with these companies does not alter our adherence to PLOS ONE policies on sharing data and materials.

# Materials and methods

## Study design

This response analysis was a planned secondary analysis of the physiological data collected during a pragmatic, randomised controlled trial comparing very early endurance and resistance training combined with mobilisation to usual care in mechanically ventilated, critically ill adults [18]. The trial was conducted in a mixed ICU of a Swiss academic centre (Department of Intensive Care Medicine, Inselspital, Bern University Hospital) between October 8, 2012 and April 5, 2016. No significant differences were found in the primary or secondary outcomes with the exception of improved mental health six months after hospital discharge for the experimental group [19]. Consequently, this secondary analysis considered the two randomised groups as one population using data from all trial participants with at least one rehabilitation session. A preliminary safety analysis of the physiological data of the first 35 subjects indicated a moderately increased workload with increased heart rate and oxygen consumption but stable oxygen saturation from before to during rehabilitation [20].

The local ethics committee approved the study that was registered in the German Clinical Trials Register (DRKS00004347) on September 10, 2012. Written informed consent was obtained from all participants or their representatives.

## Population

Participants were $\geq$ 18 years old, functionally independent before ICU admission and expected to remain ventilated for $\geq$ 72 hours. They were ineligible in cases of suspected previous muscle weakness, contraindications to cycling, palliative care, admission diagnosis that precluded walking at hospital discharge or insufficient command of German or French [18].

## Intervention delivery

Early rehabilitation interventions have been previously described [18, 19]. In brief, they started within 48 hours of ICU admission with common rehabilitation interventions provided by certified physiotherapists. Within the context of the RCT, exercise frequency, intensity, type and duration were individually tailored based on the clinical judgement of the treating physiotherapist in one group, while the other used a stepwise, standardised approach dependent upon tolerance and stability (S1 File). Patients in both groups were closely monitored and treatment interventions recorded. A pragmatic trial design was chosen to reflect real-world resources and to enhance clinical implementation. In consequence, any rehabilitation session may have included short, individually-set breaks between efforts. Physiotherapists could prematurely cease a session based on their clinical reasoning, but had to note the reason for therapy discontinuation. Adverse events were prospectively defined as 'persisting despite an intervention or therapy interruption' and included a new unstable hemodynamic, oxygen desaturation ($<85\%$), accidental fall or device dislocation [18]. This flexible approach considered individually set limits instead of strict target numbers which might be too rigid for this heterogenous population. Finally, all participants had a resting period of 30 minutes before and 15 minutes after each physiotherapy session, where they should not be disturbed.

## Data collection and measurements

Physiological data were collected by standard ICU monitoring for heart rate (HR [beats per minute: bpm]; electrocardiogram), mean arterial pressure (MAP [mmHg]; arterial line), minute ventilation (MV [l/min]; Serv0-i V3.0, Maquet Getinge Group, Gossau, Switzerland), oxygen saturation (SpO$_2$ [%]; pulse oximetry) and by indirect calorimetry for oxygen

consumption (VO$_2$ [ml/min]; CARESCAPE patient monitor B850 with E-COVX-00 Module, GE Healthcare, Finland). Indirect calorimetry was installed for planned sessions in mechanically ventilated patients from Monday to Friday who did not have a contraindication such as high FiO$_2$ (>60%), respiratory rate (>30 bpm) or intolerance for airway leak (e.g., PEEP >15 cmH$_2$O). On days with more than one physiotherapy session (<10%), only one, nonspecific session was monitored by indirect calorimetry.

All values were delivered to our patient data management system (PDMS: Centricity Critical Care Clinisoft, GE, Barrington, IL, USA) that recorded 2-minute medians to remove artefacts. We further screened and deleted isolated values that were out of range (HR ≥ 220bpm; MAP negative or ≥ 200mmHg; VO$_2$ ≤ 30ml/min). Afterwards we collected the median, maximum, minimum, mean and coefficient of variability (CV: defined as maximum minus minimum divided by mean multiplied by 100 –calculated to estimate fluctuations) for all values in the three prespecified timepoints: baseline (30min before), training (physiotherapy duration) and recovery (15min after). After each session, the treating physiotherapists recorded the time for beginning and end of physiotherapy, types of interventions, treatment modality (active, passive, mixed patient participation), mobilisation level (in-bed, edge-of-bed, out-of-bed), discontinuation or adverse events while a study nurse noted patient baseline characteristics, airway management, daily medications and Sequential Organ Failure Assessment (SOFA). Within the context of the original pragmatic randomised trial, we chose not to analyse the physiological response to isolated types of interventions, but investigated common treatment packages in the critically ill. Based on the treatment interventions performed we coded each session into seven predefined categories. These categories–termed 'session type'—were cycling, mobilisation, respiratory management, exercise, exercise and respiratory management, complex exercise and mobilisation (meaning multifaceted rehabilitation that included mobilisation), and complex cycling and mobilisation (meaning multifaceted rehabilitation that included mobilisation and cycling) (S1 File).

## Statistical analysis

**Descriptive analysis.** We describe the study population in terms of counts (n), percentages (%), means with standard deviation (SD) and medians with interquartile range (IQR). We visualized physiological trajectories over the three time-points and calculated correlations between physiological values.

**Cardiorespiratory response.** The impact of explanatory variables on physiological values during (training) and after (recovery) rehabilitation was investigated in respect to before the physiotherapy session and variability (CV) with Gaussian mixed-effects models which accounted for correlation of within-individual measurements (S1 File) based on Vickers et al. [21]. We excluded physiological values from HR and MAP with a corresponding zero CV to account for cardiac pacing. Explanatory variables were prospectively determined using extensive clinical reasoning and previous evidence to account for confounders. They included patient characteristics (age, gender, Body Mass Index (BMI), daily SOFA), attributes of the physiotherapy session (session type, mobilisation level, treatment modality, session duration, time since ICU admission), and ICU environment (airway, daily medication). We report p-values from a likelihood test to test the overall significance of categorical variables. The correlation between explanatory variables was examined to avoid collinearity (without taking into account the correlations among explanatory variables of the same subjects). The estimated coefficients may be interpreted as the impact of the explanatory variable on the change. Accordingly, these estimates characterise the impact of the explanatory variables on the average values of 'during' or 'after' rehabilitation in respect to the value 'before' rehabilitation.

**Safety analysis.** We used a mixed-effects logistic regression model which accounted for correlation of within-individual measurements to investigate explanatory variables related to clinically relevant variations and report odds ratios with 95% confidence intervals (CI). The safety cut-off for a clinically relevant variation in physiological measurements was defined as >10% variation–based on half of what is commonly reported as an adverse event (>20%) in the literature [22]. Exercise is structured and repetitive physical activity that results in energy expenditure [23]. Thus, a 10% threshold might be a safe cardiorespiratory training intensity in the critically ill adult.

Considering the exploratory, hypothesis-generating purpose of our secondary analysis, the significance threshold was set to 0.05 without adjustment for multiple testing. The statistical analysis was performed with R version 4.0.2 (2020-06-22).

## Results

During the trial, a total of 784 physiotherapy-led rehabilitation sessions were conducted in 113 participants. The duration of physiotherapy was missing for 15 sessions, which were subsequently excluded. Thus, a total of 769 sessions for 113 participants were available. After the removal of sessions with a CV of zero for HR and MAP, we analysed physiological data from 716 sessions for 108 participants with a median of 3 [2–8] sessions per subject (S1 Fig). The characteristics of these sessions and participants are described in Table 1. Numerical physiological values and their correlations from before, during and after the sessions are described in S2 File (S1 Table, S2 Fig). The strongest correlations (r > 0.5) occurred within the same physiological value between the three timepoints and between MV and $VO_2$ over all timepoints.

### Descriptive analysis

The differences between the timepoints from before to during (training) and before to after (recovery) for medians of physiological values are illustrated in Fig 1. On average, these differences were not clinically relevant although wide 95% CI indicate fluctuations across sessions (S2 File: S1 Table). Clinically relevant variations (>10%) were highest for MV (training: 35.6% of sessions, recovery: 33.8%), $VO_2$ (26.0%, 26.1%) and MAP (21.8%, 28.2%) (S2 File: S2 Table). These fluctuations become more apparent when plotting physiological trajectories across the three timepoints according to 'session type' (Fig 2). Differences for CV were equal across the three timepoints, but again varied in regards to 'session type' (S2 File: S3 and S4 Figs).

### Cardiorespiratory responses

Explanatory variables had a low correlation with themselves and were all kept in the analysis (S2 File: S2 Fig). We report the estimated effect of explanatory variables for HR, MAP, MV and $VO_2$ 'during' (Table 2) and 'after' physiotherapy-led rehabilitation (Table 3). Non-modifiable explanatory variables that mostly affected physiological responses during and after physiotherapy were age, gender, BMI and the daily SOFA score, whereas modifiable explanatory variables were session type, treatment modality, session duration, mobilisation level and daily medication. Shorter 'session duration' and 'active treatment modality' generally increased physiological parameters during rehabilitation, while cardiorespiratory parameters did not return back to baseline for 'session type' and 'mobilisation level' during the prespecified 15-min recovery-phase. Explanatory variables with an effect on $SpO_2$ were 'mobilisation level' and 'airway support' (S2 File: S4 Table).

**Table 1. Participant and physiotherapy session characteristics.**

| Participant variables | N | Median [25%, 75% quantile] or counts (%) |
|---|---|---|
| Age (years) | 108 | 66.7 [55.1, 74.4] |
| Gender (male) | 108 | 72 (67%) |
| Body Mass Index (kg/m$^2$) | 108 | 26.4 [23.7, 29.7] |
| APACHE II score (0–71) | 108 | 22.0 [17.8, 27.2] |
| SOFA score at ICU admission (0–24) | 108 | 8 [6–11] |
| ICU days until study inclusion | 108 | 1.8 [0.9–2.6] |
| ICU length of stay (days) | 108 | 7.0 [4.6–13.9] |
| Physiotherapy sessions per subject | 108 | 3 [2–8] |
| Randomised to the experimental group | 716 | 372 (52%) |
| Time from ICU admission to start of first session (days) | 108 | 2.0 [1.4–3.1] |
| Time from ICU admission to start of individual session (days) [a] | 716 | 8.6 [3.9, 19.6] |
| Physiotherapy session duration (min) | 716 | 22.0 [16.8, 30.0] |
| Daily SOFA score during physiotherapy (0–24) | 713 | 8 [5, 12] |
| Session types | 716 | |
| exercise | | 193 (27%) |
| cycling | | 160 (22%) |
| mobilisation | | 178 (25%) |
| respiratory management | | 66 (9%) |
| exercise and respiratory management | | 54 (8%) |
| complex cycling and mobilisation | | 10 (1%) |
| complex exercise and mobilisation | | 55 (8%) |
| Treatment modality | 610 | |
| passive | | 401 (66%) |
| active | | 187 (31%) |
| mixed | | 22 (4%) |
| Mobilisation level | 716 | |
| in-bed | | 488 (68%) |
| edge-of-bed | | 150 (21%) |
| out-of-bed | | 78 (11%) |
| Physiotherapy session discontinuation [b] | 716 | 23 (3%) |
| Adverse event during physiotherapy | 716 | 4 (1%) |
| Airway support | 716 | |
| endotracheal tube | | 327 (46%) |
| tracheostomy | | 271 (38%) |
| none | | 118 (16%) |
| Neuromuscular blocking agents on day of session | 716 | 98 (14%) |
| Vasoactive support on day of session | 716 | 371 (52%) |
| Opiates on day of session | 716 | 662 (92%) |
| Sedatives on day of session | 716 | 539 (75%) |

[a] number of sessions varied between patients, this variable takes into account the time from ICU admission to the start of each, individual session in the individual patient.

[b] for all physiotherapy sessions (n = 784) there were 25 (3%) therapy discontinuations (two sessions with a therapy discontinuation from one subject were excluded in this analysis because of zero CV).

**Abbreviations**: SD = standard deviation, APACHE = Acute Physiology and Chronic Health Evaluation, SOFA = Sequential Organ Failure Assessment.

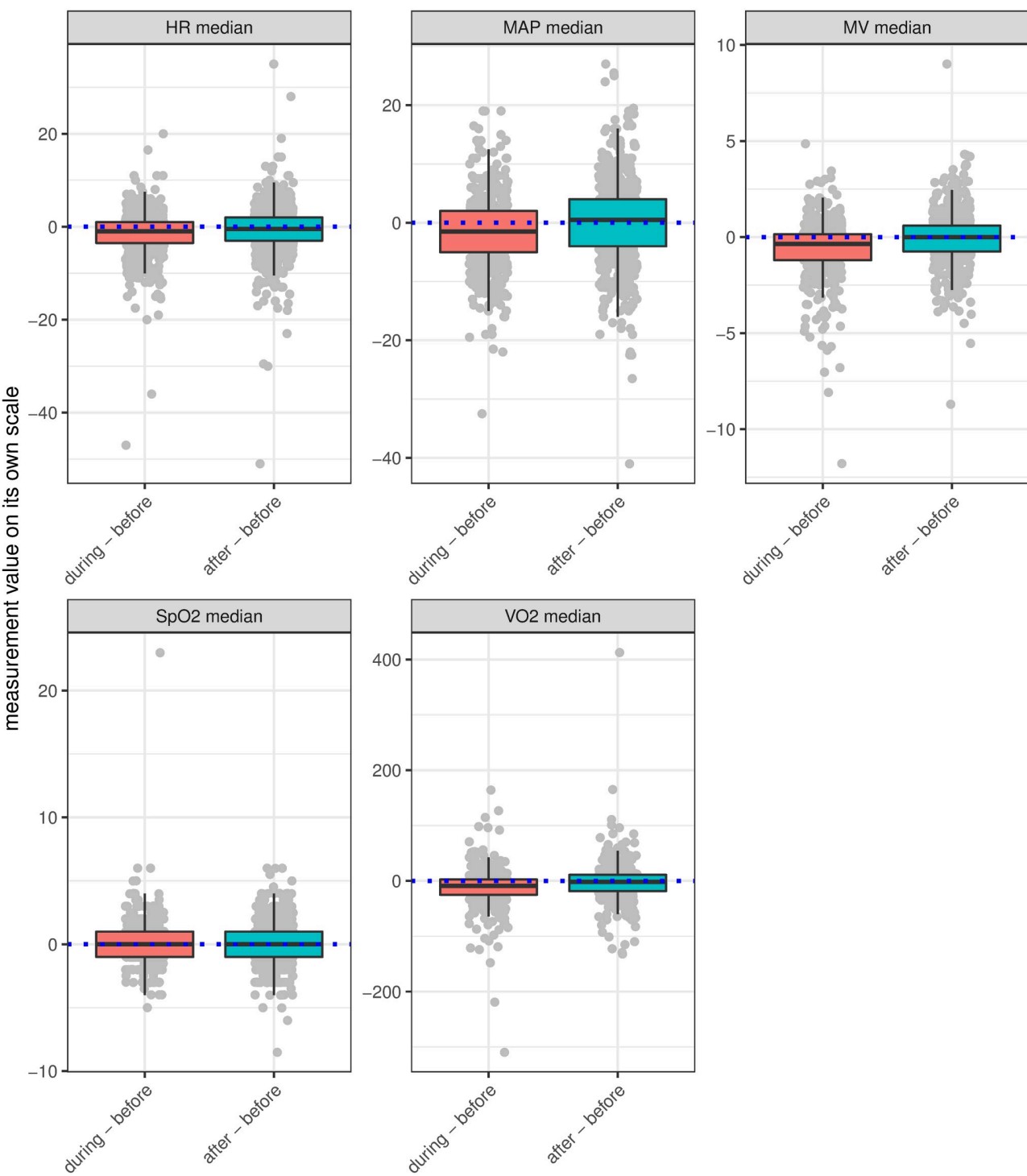

**Fig 1. Differences in median physiological values from before to during/after physiotherapy.** Median differences were calculated as "during minus before" (training) and "after minus before" (recovery) physiotherapy for all physiological values. Measurement units: HR (bpm), MAP (mmHg), MV (l/min), SpO$_2$ (%), VO$_2$ (ml/min). Numerical differences (median [25%, 75%]): HR: training 1bpm [-1, 3.5], recovery 0.5bpm [−2, 3]; MAP: training 1.5mmHg [−2, 5], recovery -0.5 mmHg [−4, 4]; MV training 0.35l/min [-0.15, 1.2], recovery 0l/min [-0.6, 0.75]; SpO$_2$: training 0% [−1, 1], recovery 0% [−1, 1]; VO2 training 8.8ml/min [-2.47, 25], recovery 1.65ml/min [-10.9, 18.3] (S2 File: S1 Table).

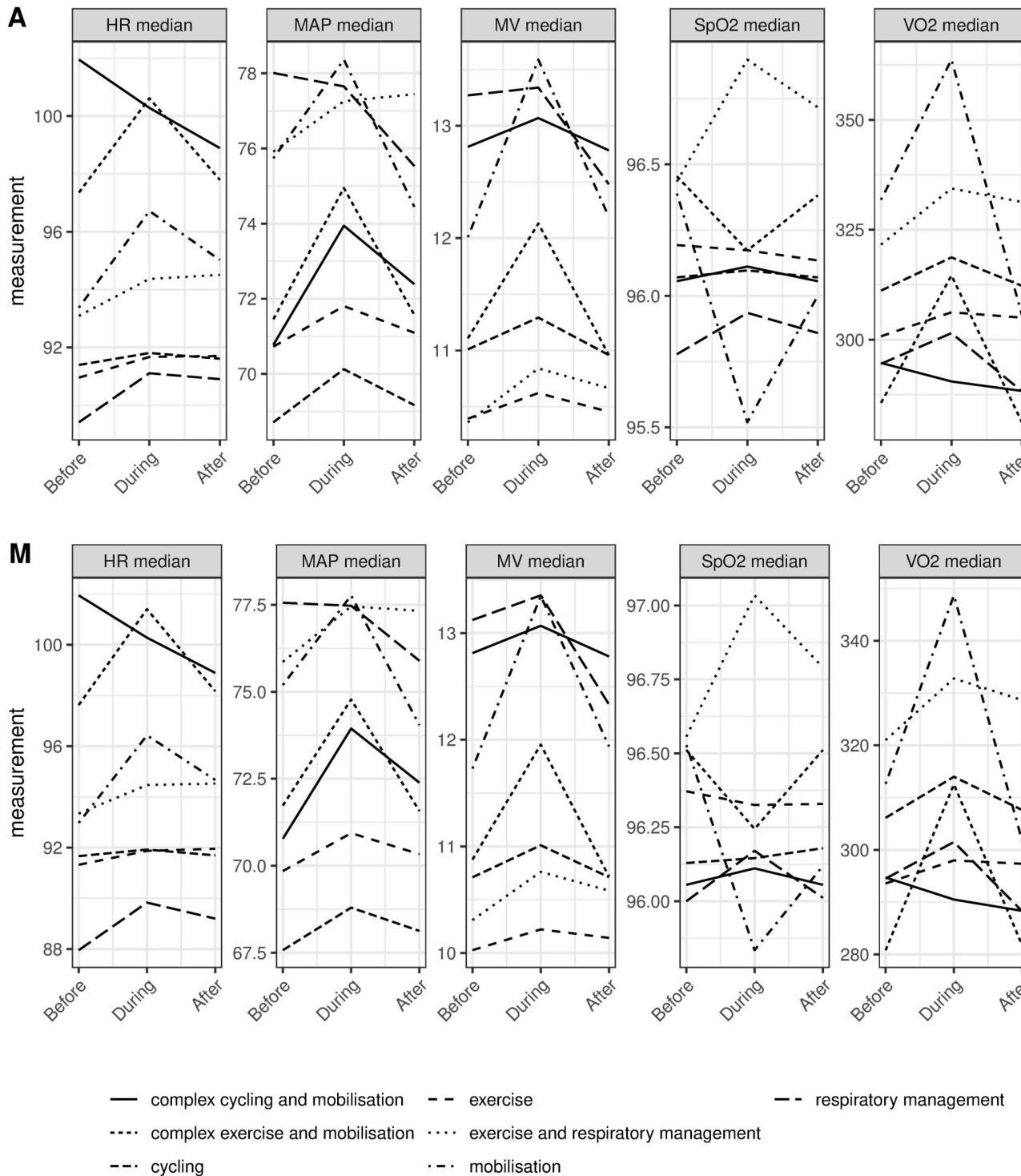

**Fig 2. Trajectories of median physiological values according to session type.** A) Average trajectories with standard errors. M) median trajectories with IQR of physiological measurements. Standard errors are computed under the assumption of independences among all the observations. Measurement units: HR (bpm), MAP (mmHg), MV (l/min), SpO2 (%), VO2 (ml/min).

**Table 2. Estimated fixed effect of explanatory variables on HR, MAP, MV and VO$_2$ 'during' rehabilitation.**

| Explanatory variables | HR during (95%-CI) | MAP during (95%-CI) | MV during (95%-CI) | VO$_2$ during (95%-CI) |
|---|---|---|---|---|
| Number of sessions | 571 | 535 | 442 | 312 |
| **Age (years)** [a] | 0.02 (-0.02, 0.06) | 0.01 (-0.03, 0.05) | -0.002 (-0.01, 0.01) | **-0.54 (-0.87, -0.21)** |
| **Gender (male is reference)** | 0.24 (-0.84, 1.32) | **1.03 (0.01, 2.05)** | **-0.42 (-0.72, -0.12)** | **-23.78 (-32.74, -15.39)** |
| Body Mass Index (kg/m$^2$) [a] | 0.02 (-0.09, 0.12) | 0.04 (-0.07, 0.14) | -0.01 (-0.04, 0.02) | 0.88 (-0.02, 1.77) |
| Daily SOFA score (0–24) [a] | 0.002 (-0.11, 0.12) | -0.02 (-0.15, 0.11) | 0.01 (-0.02, 0.05) | 0.29 (-0.73, 1.32) |
| **Session duration (min)** [a] | **0.05 (0.01, 0.09)** | **-0.06 (-0.12, -0.01)** | **-0.02 (-0.03, -0.00)** | **-0.59 (-1.10, -0.13)** |
| Time from ICU admission to start of session (days) [a] | 0.02 (-0.01, 0.06) | 0.03 (-0.02, 0.07) | 0.001 (-0.01, 0.01) | -0.18 (-0.59, 0.20) |
| Session type (exercise is reference) | | | | |
| p-value [b] | 0.136 | 0.627 | 0.434 | 0.615 |
| cycling | -0.93 (-2.05, 0.24) | 0.45 (-0.94, 1.84) | 0.15 (-0.17, 0.48) | 3.52 (-6.38, 14.18) |
| mobilisation | 2.05 (-1.79, 5.92) | 0.28 (-4.68, 5.25) | 0.79 (-0.36, 1.93) | 38.54 (-14.80, 89.51) |
| respiratory management | 0.72 (-0.92, 2.40) | 1.83 (-0.33, 4.00) | 0.36 (-0.36, 1.14) | 1.56 (-62.48, 65.85) |
| exercise and respiratory management | 0.29 (-1.15, 1.70) | 0.25 (-1.55, 2.06) | -0.02 (-0.46, 0.43) | -0.04 (-14.13, 14.72) |
| complex cycling and mobilisation | -2.59 (-5.74, 0.54) | 1.98 (-1.99, 5.95) | 0.19 (-0.70, 1.08) | -7.51 (-35.63, 20.44) |
| complex exercise and mobilisation | 2.21 (-1.50, 5.98) | 1.28 (-3.53, 6.08) | 0.24 (-0.86, 1.33) | 19.71 (-27.29, 66.16) |
| **Treatment modality (passive is reference)** | | | | |
| p-value [b] | 0.064 | **0.016** | **<0.001** | **<0.001** |
| **mixed** | -0.38 (-2.35, 1.59) | **2.88 (0.26, 5.51)** | 0.53 (-0.06, 1.12) | 1.11 (-17.97, 20.88) |
| **active** | **1.16 (0.13, 2.19)** | **1.41 (0.13, 2.69)** | **0.72 (0.40, 1.04)** | **23.01 (12.51, 33.97)** |
| Mobilisation level (in-bed is reference) | | | | |
| p-value [b] | 0.610 | 0.900 | 0.294 | 0.681 |
| edge-of-bed | -0.78 (-4.58, 3.03) | 0.88 (-4.01, 5.78) | 0.74 (-0.41, 1.88) | -2.85 (-52.04, 47.78) |
| out-of-bed | 0.13 (-3.91, 4.08) | 0.81 (-4.36, 5.99) | 0.36 (-0.84, 1.55) | 6.87 (-45.68, 63.19) |
| Airway support (none is reference) | | | | |
| p-value [b] | 0.502 | 0.382 | 0.379 | 0.988 |
| tracheostomy | -0.90 (-2.36, 0.59) | 0.94 (-0.89, 2.77) | 0.30 (-0.33, 0.96) | 1.36 (-24.32, 27.77) |
| endotracheal tube | -0.49 (-1.77, 0.83) | 1.14 (-0.52, 2.81) | 0.05 (-0.54, 0.68) | 0.85 (-23.42, 25.36) |
| Opiates on session day [c] | 0.39 (-1.10, 1.81) | 0.62 (-1.41, 2.64) | 0.12 (-0.44, 0.67) | -7.44 (-24.94, 9.70) |
| **Vasoactive on session day** [c] | -0.11 (-1.04, 0.81) | **-1.79 (-2.95, -0.64)** | -0.13 (-0.41, 0.15) | -6.92 (-15.76, 2.16) |
| **Sedatives on session day** [c] | **1.15 (0.06, 2.20)** | 1.32 (-0.05, 2.68) | 0.04 (-0.34, 0.41) | 4.88 (-8.80, 19.34) |
| Neuromuscular blocking agents on session day [c] | -0.19 (-1.25, 0.89) | -0.37 (-1.73, 0.98) | -0.21 (-0.51, 0.11) | -6.73 (-16.85, 3.20) |

[a] per one-unit increase (for continuous variables).

[b] likelihood test for overall significance of categorical variables.

[c] none is reference.

Reported effects of explanatory variables are mean differences of median values and need to be considered under the assumption of 'all other covariates being constant'. All models are adjusted for measured values and CV before physiotherapy (estimates not shown). Examples for interpretation: **(1) Categorical data**: Median VO$_2$ significantly increased during training by 23.01ml/min with *active* patient participation when compared to *passive* patient participation. **(2) Continuous data**: Median VO$_2$ decreased during training per one additional year of age by 0.54ml/min.

## Safety analysis

There were a total of 27 sessions where either an adverse event (4 [0.6%]) or a discontinuation of rehabilitation (23 [3%]) occurred (Table 1). Events and participants' characteristics are described in S3 File. Overall, there were too few adverse and discontinuation events to investigate their patterns or predictors. We did, however, investigate factors for clinically relevant variations along with the two outliers in Fig 1. Thereby, 'mobilisation level', 'treatment modality' and 'session type' were most likely to induce clinically relevant variations. For example,

**Table 3. Estimated fixed effect of explanatory variables on HR, MAP, MV and VO$_2$ 'after' rehabilitation.**

| Explanatory variables | HR after (95%-CI) | MAP after (95%-CI) | MV after (95%-CI) | VO$_2$ after (95%-CI) |
|---|---|---|---|---|
| Number of sessions | 569 | 534 | 437 | 308 |
| **Age (years)** [a] | **0.05 (0.01, 0.08)** | 0.001 (-0.05, 0.05) | 0.002 (-0.01, 0.01) | -0.30 (-0.68, 0.08) |
| **Gender (male is reference)** | 0.47 (-0.39, 1.52) | 0.50 (-0.75, 1.74) | **-0.29 (-0.58, -0.01)** | **-18.23 (-27.91, -8.55)** |
| **Body Mass Index (kg/m$^2$)** [a] | -0.06 (-0.15, 0.04) | **-0.13 (-0.26, -0.00)** | -0.02 (-0.05, 0.01) | 0.52 (-0.51, 1.55) |
| **Daily SOFA score (0–24)** [a] | **-0.13 (-0.24, -0.01)** | **-0.20 (-0.36, -0.04)** | -0.01 (-0.04, 0.03) | -0.32 (-1.50, 0.86) |
| Session duration (min) [a] | 0.01 (-0.04, 0.06) | -0.01 (-0.07, 0.05) | -0.0002 (-0.02, 0.01) | -0.28 (-0.86, 0.30) |
| Time from ICU admission to start of session (days) [a] | 0.03 (-0.01, 0.07) | 0.01 (-0.04, 0.06) | -0.003 (-0.02, 0.01) | 0.08 (-0.38, 0.53) |
| **Session type (exercise is reference)** | | | | |
| p-value [b] | **0.016** | 0.916 | **0.048** | 0.982 |
| cycling | -0.53 (-1.73, 0.77) | -0.37 (-2.06, 1.33) | -0.04 (-0.39, 0.31) | -2.38 (-14.13, 9.37) |
| **mobilisation** | **6.56 (2.14, 11.24)** | -1.78 (-7.82, 4.27) | 1.16 (-0.13, 2.45) | -12.17 (-73.47, 49.13) |
| respiratory management | 0.55 (-1.42, 2.40) | -0.69 (-3.33, 1.94) | -0.81 (-1.66, 0.05) | -15.98 (-90.71, 58.75) |
| exercise and respiratory management | 0.12 (-1.64, 1.67) | 0.32 (-1.87, 2.52) | 0.07 (-0.43, 0.57) | -4.17 (-21.07, 12.73) |
| complex cycling and mobilisation | -2.79 (-6.37, 0.96) | 0.71 (-4.12, 5.53) | 0.18 (-0.81, 1.17) | -11.64 (-44.25, 20.96) |
| **complex exercise and mobilisation** | **4.52 (0.22, 9.07)** | -0.35 (-6.19, 5.50) | 0.18 (-1.05, 1.41) | -12.63 (-67.08, 41.82) |
| Treatment modality (passive is reference) | | | | |
| p-value [b] | 0.167 | 0.178 | 0.181 | 0.095 |
| mixed | -0.95 (-3.33, 1.30) | 0.18 (-3.02, 3.37) | -0.43 (-1.10, 0.24) | -6.55 (-29.10, 15.99) |
| active | 0.97 (-0.27, 2.07) | 1.44 (-0.13, 3.00) | 0.19 (-0.16, 0.55) | 11.98 (-0.52, 24.48) |
| **Mobilisation level (in-bed is reference)** | | | | |
| p-value [b] | **0.027** | 0.578 | **0.050** | **0.001** |
| edge-of-bed | **-5.85 (-10.43, -1.42)** | -1.35 (-7.30, 4.61) | -0.39 (-1.67, 0.89) | 13.41 (-44.79, 71.60) |
| out-of-bed | **-6.06 (-11.05, -1.67)** | -2.62 (-8.92, 3.68) | -1.18 (-2.51, 0.16) | -40.92 (-104.37, 22.52) |
| Airway support (none is reference) | | | | |
| p-value [b] | 0.416 | 0.471 | 0.678 | 0.886 |
| tracheostomy | 0.11 (-1.48, 1.76) | 0.50 (-1.73, 2.74) | 0.30 (-0.42, 1.02) | 4.88 (-25.36, 35.11) |
| endotracheal tube | 0.87 (-0.66, 2.33) | 1.16 (-0.86, 3.19) | 0.18 (-0.51, 0.87) | 6.34 (-22.03, 34.70) |
| Opiates on session day [c] | 0.43 (-1.25, 2.08) | -0.06 (-2.56, 2.43) | 0.35 (-0.28, 0.97) | -8.27 (-28.25, 11.72) |
| **Vasoactive on session day** [c] | 0.26 (-0.78, 1.33) | **-2.42 (-3.82, -1.01)** | -0.28 (-0.59, 0.03) | **-11.78 (-22.17, -1.39)** |
| **Sedatives on session day** [c] | **1.32 (0.00, 2.44)** | 1.03 (-0.64, 2.69) | 0.24 (-0.18, 0.66) | 10.86 (-5.72, 27.43) |
| Neuromuscular blocking agents on session day [c] | 0.18 (-1.04, 1.45) | -0.08 (-1.73, 1.57) | -0.01 (-0.35, 0.34) | -10.12 (-21.78, 1.54) |

[a] per one-unit increase (for continuous variables).

[b] likelihood test for overall significance of categorical variables.

[c] none is reference.

Reported effects of explanatory variables are mean differences of median values and need to be considered under the assumption of 'all other covariates being constant'. All models are adjusted for measured values and CV before physiotherapy (estimates not shown). Examples for interpretation: **(1) Categorical data**: Median HR 'after' (recovery) significantly increased for the 'session type' *mobilisation* with 6.56pbm when compared to *exercise*. Co-occurrence of 'mobilisation levels' and 'session types' are reported in S2 File (S3 Table E3). **(2) Continuous data**: A one-point increase in 'SOFA score' significantly reduced median MAP during recovery by 0.2mmHg.

patients with an out-of-bed/edge-of-bed mobilisation had a 2.3 times higher chance (odds ratio 95%CI 0.4–13.4, p = 0.36) to demonstrate a clinically relevant 10% variations in 'VO$_2$ during' and a 4.7 times higher chance (95%CI 0.46–48.42, p = 0.19) in 'VO$_2$ after' compared to an in-bed mobilisation (S3 File).

## Discussion

This study provides new insight into the cardiorespiratory response to early rehabilitation in mechanically ventilated, critically ill adults in a mixed ICU. The main findings are:

1. Physiological values had a large range of variation during rehabilitation and recovery within and across participants' sessions. Clinically relevant variations occurred in a substantial number of patients, despite overall changes not being statistically significant in the whole group of patients.

2. Key modifiable explanatory variables for physiological changes were session type, treatment modality, session duration, mobilisation level and daily medication, while key non-modifiable explanatory variables were age, gender, BMI and the SOFA score on the day of physiotherapy. Specifically, clinicians should be aware that during rehabilitation cardiorespiratory parameters (HR, MAP, MV and $VO_2$) increase when using active participation versus passive therapy and decrease (with the exception of HR) with each additional minute of rehabilitation. In contrast, HR remains elevated in the recovery phase after (multimodal) mobilisation when compared to exercise and drops following higher mobilisation levels when compared to in-bed rehabilitation.

The findings from our study enhance clinical reasoning and decision-making about the beginning, type, duration and intensity of physiotherapy-led rehabilitation by informing clinicians about the estimated effects on physiological values based on individual patient characteristics. For example, we found no evidence that the 'time from ICU admission to the start of physiotherapy' particularly affected physiological values. Instead 'session type', 'treatment modality' and 'session duration' seem to be the major drivers of cardiorespiratory changes and should be used to plan appropriate rehabilitation interventions. Given the good correlation between $VO_2$ and MV, we recommend to monitor routine data such as HR, MAP, MV and $SpO_2$ and to tailor training intensity as well as to ensure sufficient recovery.

The goal of rehabilitation is to optimise physical functioning in order to enhance autonomy and participation [24]. Physiological instability is considered a barrier to the safe implementation of early rehabilitation in the ICU [25]. Our partly contradictory results–no general effect, despite large variations–might mirror the challenge of providing safe rehabilitation within critical care. Therapists need to balance safety against a sufficient training stimulation. We found that a clinically relevant variation was achieved in 1 out of 4 sessions. The physiological reaction of critically ill patients can vary from day to day within and across participants as indicated by the high variation and large confidence intervals of physiological values. Early rehabilitation should therefore be closely monitored and individually tailored, whereby changes in physiological parameters might necessitate adaptions within a session. Clearly, early rehabilitation is not a 'one size fits all' approach but rather requires continuous clinical reasoning and interprofessional collaboration.

The finding that per additional minute of rehabilitation cardiorespiratory parameters decreased is interesting. A potential explanation is that shorter sessions were more intensive with less breaks in-between. Alternatively, repeated muscle activation might be limited in the critically ill leading to early onset of muscle fatigue and patient inactivity [26]. From a training perspective, shorter and more frequent sessions might therefore be preferred to achieve an adequate training response. This strategy was associated with improved 3-month outcomes after stroke [27] and should be investigated in future randomised controlled trials.

Our results further reveal differences in the cardiorespiratory response of males versus females, elderly patients (>63 years), who may not achieve a sufficient cardiorespiratory response to training anymore and thus may require longer time for recovery, or of patients with a higher SOFA score possibly due to bioenergetic dysfunction [17]. There are few studies specifically investigating the impact of 'illness severity' on the cardiorespiratory response to physiotherapy. One retrospective study (n = 23) examining early mobilisation in the elderly (>75 years, APACHE II: 27) did not find any adverse reactions in haemodynamic parameters

[28]. Another trial investigating early cycling in adults with septic shock (n = 18) suggests that this intervention is safe (0.4% adverse events) and might preserve muscle cross-sectional areas, though authors do not report haemodynamic data to support this [29]. Finally, a recent study testing early graded, passive cycling in septic patients (n = 10, SOFA = 7.5) similarly found a high variation that ranged from improved to worsened left ventricular function between patients [30].

In our study, physiotherapy in patients with vasoactive support seemed safe, though drops in MAP were likely during both training and recovery. Rebel et al. [31] similarly found that mobilisation with vasoactive support is feasible and safe, but associated with a higher risk of hypotension. The risk of hypotension in our study seems higher in obese (BMI >27), sicker (SOFA ≥8) patients, with 'vasoactive support' or longer 'session duration' (>24min), but not with 'session type'. Previous findings about passive exercise or cycling with or without vasoactive support not affecting haemodynamic parameters [12, 32], could therefore be due to their case-mix. Our results furthermore indicate that HR might slightly increase during and after rehabilitation in patients receiving sedatives. Clinicians need to be aware of such associations and adjust exercise to remain within their therapeutic targets. Still, early rehabilitation could detect readiness to wean sedatives–shortening time on mechanical ventilation–and is therefore recommended by international guidelines [33].

Finally, clinicians need to be aware that heart rate recovery may be prolonged in elderly patients (>63 years) or after mobilisation when compared to just exercise. This is substantiated by the study of Black et al. [17] who found prolonged recovery times (defined as return to 10% of baseline $VO_2$) for 1 in 4 rehabilitation sessions. We also found >25% of clinically relevant variations during recovery in MAP, MV and $VO_2$ with $VO_2$ remaining slightly but not significantly elevated. It is important to note that in our data rehabilitation included the whole spectrum of interventions while Black et al. [17] specifically investigated active, out-of-bed activities in long-stayers. Our descriptive data might therefore underestimate the cardiorespiratory response to these activities and patient group, but seem more generalisable to mixed ICU patients across their whole ICU stay. Additionally, it allowed us to specifically analyse explanatory variables and their estimated effect. In this regard, our results support previous research that active exercises and out-of-bed mobilisations lead to stronger physiological reactions than passive exercises or in-bed mobilisations [11, 14–17, 34, 35]. Nevertheless, the cardiorespiratory response to 'session duration', 'session type' and 'mobilisation level' does not seem as straightforward due to considerable overlap (S2 File: S3 Table). For example, while session type 'mobilisation' was generally associated with increased HR in the recovery period, HR dropped substantially following an 'out-of-bed' mobilisation. This phenomenon might signify increased blood flow after returning to the supine position or might be a sign of exhaustion. Still, the incidence of adverse events remained very low (0.6% [5/784 physiotherapy sessions]), indicating that early rehabilitation in the critically ill, mechanically ventilated adult is safe.

This study has limitations. First, this analysis relied solely on physiological data and did not consider fatigue or other subjective measures to evaluate patients' training load or recovery. This might be important because exhaustion remains a major barrier to patient participation [36] and might not be reflected by physiological data. Second, we cannot exclude potential measurement error as we relied on standard ICU monitoring, whereby median filtering should have reduced the risk of artefacts substantially [37]. Third, the original trial primarily aimed to assess the efficacy of early rehabilitation while monitoring the safety of the randomised interventions. The population was therefore highly selective. Additionally, the estimated effect of explanatory variables should be interpreted as hypothesis-generating and in the context of safety. Yet, to the best of our knowledge, this is the first analysis that explored various explanatory variables on a large critically ill sample across the whole ICU stay. These results therefore

provide important information for future trials, but need to be validated in prospective studies. Fourth, while our results inform clinical decision-making on the intensity and duration of early rehabilitation, they cannot establish the effect on functional outcomes. Finally, the safety cut-off for a clinically relevant variation was chosen in absence of previous evidence. However, a cut-off of 20% still led to variations in 1 out of 10 sessions, particularly for MV and $VO_2$ (S2 File: S2 Table). Our interventions were safe with only few, transient adverse events. Future trials should therefore investigate the feasibility and efficacy of different physiological training intensities as well as their association with neuromuscular activation and patients' perceived rate of exertion.

## Conclusions

Based on the physiological data from 'before', 'during' and 'after' 716 early physiotherapy-led rehabilitation sessions from 108 participants, our findings indicate that rehabilitation in the ICU is safe and does not negatively influence physiological parameters. Nevertheless, clinically relevant variations are common during training and the recovery period. Physiological parameters should therefore be closely monitored and exercise individually tailored. The explanatory variables identified guide clinicians' decision-making in delivering the optimal type and intensity by enabling clinicians to estimate the cardiorespiratory response prospectively. Shorter sessions and active treatment therefore seem to increase the cardiorespiratory response during therapy, while sufficient time for recovery seems particularly necessary after a mobilisation session.

## Supporting information

**S1 File. Detailed methodology.**
(PDF)

**S2 File. Additional results including S1–S4 Tables and S2–S5 Figs.**
(PDF)

**S3 File. Adverse events, therapy discontinuation and factors for clinically relevant variations.**
(DOCX)

**S4 File. Codebook.**
(XLSX)

**S5 File. Dataset.**
(CSV)

**S6 File. Analysis code and regression output.**
(PDF)

**S7 File. STROBE checklist.**
(DOCX)

**S1 Fig. Flow diagram of rehabilitation sessions.**
(TIF)

## Author Contributions

**Conceptualization:** Sabrina Eggmann, Irina Irincheeva, Gere Luder, Martin L. Verra, Caroline H. G. Bastiaenen, Stephan M. Jakob.

**Data curation:** Sabrina Eggmann, Irina Irincheeva, André Moser.

**Formal analysis:** Sabrina Eggmann, Irina Irincheeva, Gere Luder, André Moser.

**Funding acquisition:** Sabrina Eggmann.

**Investigation:** Sabrina Eggmann, Gere Luder, Martin L. Verra, Stephan M. Jakob.

**Methodology:** Sabrina Eggmann, Irina Irincheeva, Gere Luder, Martin L. Verra, André Moser, Caroline H. G. Bastiaenen.

**Project administration:** Sabrina Eggmann.

**Resources:** Sabrina Eggmann, Stephan M. Jakob.

**Software:** Sabrina Eggmann, Irina Irincheeva, André Moser.

**Supervision:** Sabrina Eggmann, Stephan M. Jakob.

**Validation:** Sabrina Eggmann, Stephan M. Jakob.

**Visualization:** Sabrina Eggmann, Irina Irincheeva, André Moser.

**Writing – original draft:** Sabrina Eggmann.

**Writing – review & editing:** Sabrina Eggmann, Irina Irincheeva, Gere Luder, Martin L. Verra, André Moser, Caroline H. G. Bastiaenen, Stephan M. Jakob.

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
