## [Decision Letter · Decision Letter 0]

25 Oct 2021

PONE-D-21-27687Cardiorespiratory response to early rehabilitation in critically ill adults: a secondary analysis of a randomised controlled trialPLOS ONE

Dear Dr. Eggmann,

Thank you for submitting your manuscript to PLOS ONE. After careful consideration, we feel that it has merit but does not fully meet PLOS ONE’s publication criteria as it currently stands. Therefore, we invite you to submit a revised version of the manuscript that addresses the points raised during the review process.

Please pay particular attention to the analytical concerns raised by all three reviewers; carefully defining all variables and terms used (with rationale where necessary) and consistency of terminology throughout.

We look forward to receiving your revised manuscript.

Kind regards,

Brenda M. Morrow, PhD

Academic Editor

PLOS ONE

Journal Requirements:

“We have read the journal's policy and the authors of this manuscript have the following competing interests: SE, GL, MLV, CHGB declare that they have no competing interests. II and AM are affiliated with CTU Bern, University of Bern, which has a staff policy of not accepting honoraria or consultancy fees. However, CTU Bern is involved in design, conduct, or analysis of clinical studies funded by not-for-profit and for-profit organizations. In particular, pharmaceutical and medical device companies provide direct funding to some of these studies. For an up-to-date list of CTU Bern’s conflicts of interest see http://www.ctu.unibe.ch/research/declaration_of_interest/index_eng.html. SMJ reports the following potential conflicts of interest: The Department of Intensive Care Medicine has, or has had in the past, research contracts with Orion Corporation, Abbott Nutrition International, B. Braun Medical AG, CSEM SA, Edwards Lifesciences Services GmbH, Kenta Biotech Ltd, Maquet Critical Care AB, Omnicare Clinical Research AG and research & development/consulting contracts with Edwards Lifesciences SA, Maquet Critical Care AB, and Nestlé. The money was paid into a departmental fund; SMJ received no personal financial gain. The Department of Intensive Care Medicine has received unrestricted educational grants from the following organizations for organizing a quarterly postgraduate educational symposium, the Berner Forum for Intensive Care (until 2015): Fresenius Kabi, gsk, MSD, Lilly, Baxter, astellas, AstraZeneca, B | Braun, CSL Behring, Maquet, Novartis, Covidien, Nycomed, Pierre Fabre Pharma AG (formerly known as RobaPharm), Pfizer, Orion Pharma, Bard Medica S.A., Abbott AG, Anandic Medical Systems. The Department of Intensive Care Medicine has received unrestricted educational grants from the following organizations for organizing bi-annual postgraduate courses in the fields of critical care ultrasound, management of ECMO and mechanical ventilation: Pierre Fabre Pharma AG (formerly known as RobaPharm), Pfizer AG, Bard Medica S.A., Abbott AG, Anandic Medical Systems, PanGas AG Healthcare, Orion Pharma, Bracco, Edwards Lifesciences AG, Hamilton Medical AG, Fresenius Kabi (Schweiz) AG, Getinge Group Maquet AG, Dräger Schweiz AG, Teleflex Medical GmbH. None of these companies had a role in this study.”

Reviewers' comments:

Reviewer's Responses to Questions

**Comments to the Author**

1. Is the manuscript technically sound, and do the data support the conclusions?

Reviewer #1: Yes

Reviewer #2: Yes

Reviewer #3: Partly

2. Has the statistical analysis been performed appropriately and rigorously? 

Reviewer #1: Yes

Reviewer #2: Yes

Reviewer #3: No

3. Have the authors made all data underlying the findings in their manuscript fully available?

Reviewer #1: Yes

Reviewer #2: Yes

Reviewer #3: Yes

4. Is the manuscript presented in an intelligible fashion and written in standard English?

Reviewer #1: Yes

Reviewer #2: Yes

Reviewer #3: Yes

5. Review Comments to the Author

Reviewer #1: Dear authors,

thank you very much for the opportunity to review the submitted manuscript. The authors conducted a secondary analysis of 108 critical care patients and analyzed the cardiorespiratory responses to rehabilitation exercises.

I am appreciating this work and have got some minor concerns.

SERIOUS CONCERNS

MINOR CONCERNS

Page 2, Line 30-33: wording: you used the terms “early rehabilitation”, “exercise”, “rehabilitation”, “training” and “recovery” as synonyms, I guess to improve readability. With a closer look, all those terms are different concepts, requiring different scientific approaches. Similar, I found in section introduction (p4, L105-110), that you used the term “early physiotherapy-led rehabilitation” here, but also “early mobilisation”, “exercise”, and “sessions”. At all, I suggest using one (1) term stringently.

P2, L37: would you please add the information of the seven different session types, with increasing activities. Please use a stringent reporting.

P2, Abstract: I am fan of clear structures, e.g. using always the same order of reported outcome parameters: MV, VO2, MAP. Within the sentences, this order changes. Would you please re-arrange the order, and keep it? See also lines 153-156.

P2, Abstract: I guess all results are mean differences? If so, please add this information.

P3, keywords: to improve your future citation, please check for pubmed’s MeSH terms (https://www.ncbi.nlm.nih.gov/mesh) for the used keywords, and schedule key words in alphabetically order.

L123: this was an RCT, but you analyse intervention- and control group as one single group? If so, please add some information about the intervention and an explanation why the summary is reasonable (if I remember right, the intervention group had a little bit more minutes of rehab each day, right?)

L125 “physiological data (n=35)” You don’t mean that you analyzed 35 data, but data of 35 patients, right?

L126: please insert a line break between “…physiotherapy [20].” and “The local…”

L130: please add “Population”, and a line break. I guess you mean “equal or older than 18 years”?

Section results is fine. The tables are hard to read, but I have no idea how to improve readability. The “examples for interpretation” are really helpful.

L316-319: would you please summarize the main results in their plain meaning, eg. “complex exercise and mobilisation led to increased VO2”, or else

Section discussion is fine and interesting.

L339, 346, 358, 369, 388: any new hypothesis or research questions?

Not part of the review and just a suggestion: You started very early, at day 2, and the muscle loss might not be severe in early days. I wonder if there are any differences in cardiorespiratory responses to PT’s rehab when delivered within early <7 days or late ≥7 days?

Very fine work, thank you.

Reviewer #2: Thank you for the opportunity to review this paper. The authors present the cardiorespiratory response in critically ill adult patients in response to various early rehabilitation interventions. This paper is well written and contributes novel data to the field, highlighting that early rehabilitation is safe.

The authors adequately identify limitations to the study. The heterogeneity in both the population and interventions and the impact on the conclusions is mentioned.

I have detailed a few minor points below:

1. It may be valuable to clarify/better define the "explanatory" variables. How were these decided on or chosen?

2. If understood correctly, the clinically relevant cutoff of >10% is related to safety; however further justification as to why this was chosen and not 20% is required. This will perhaps be important when weighing up safety versus adequate intensity to bring about physiological change. In the discussion it seems as if this clinically relevant response is linked to response to exercise and not safety – please clarify.

3. Stronger motivation is needed for not adjusting for multiple testing is needed.

4. In Table 1, please could you clarify what is meant by "Time from ICU admission to start of each session (days)"?

5. While recommendations are made for clinical practice in terms of monitoring and recovery, what may be more helpful to clinicians and advancing the cause of early rehabilitation in the ICU is know how these physiological responses link to various outcomes. As highlighted by the authors, there is a gap in terms of adequate prescription (FITT) in the critically ill patients. Such data sets, could help shed light on factors such as the frequency and intensity of rehabilitation interventions to effect change on outcomes.

Authors highlight these results enhance clinical decision making around frequency and intensity, however, I think without looking at outcomes – this enhancement is limited – it may be safe, but there is no way to know it is effective with the data provided.

Reviewer #3: This manuscript is a straightforward secondary analysis of data generated from a randomized controlled trial, investigating the effect of explanatory variables on physiological changes during training and recovery. While the study (and the analysis) looks timely, relevant, and on target, I state my thoughts below, primarily on the statistical analysis presented.

1. A multi-level logistic regression was proposed. However, it was not clear from the writeup (Statistical Analysis section) on how exactly the dichotomization of the response variable was considered, and how the multi-level part was handled? Was it via a generalized estimating equations, or via some mixed-effects models? A clear writeup is expected.

2. Tables 2 and 3 summarizes the results (estimated fixed effects of explanatory variables); it was not clear whether the table entries (estimates) are the log(oods-ratios), or something else? This needs to be made clear in the Table captions.

3. How about assessing the goodness-of-fit (GOF) after these multi-level logistic regression fits? While the Hosmer-Lemeshow is ideal for assessing the standard logistic regression fit, there exists various proposals for the multi-level setup; see below:

The authors may consider producing some summary GOF statistics in this context.

https://digscholarship.unco.edu/cgi/viewcontent.cgi?article=1243&context=dissertations

4. Interpretation of covariates/explanatory variables in the Results section should be considered in terms of increase/decrease in odds, with associated 95% confidence intervals.

6. PLOS authors have the option to publish the peer review history of their article (what does this mean?). If published, this will include your full peer review and any attached files.

Reviewer #1: **Yes: **Dr. Peter Nydahl

Reviewer #2: No

Reviewer #3: No

---

## [Author Response · Author response to Decision Letter 0]

24 Nov 2021

Responses to Reviewers

The reviewers’ comments are shown in bold type, our responses in plain type and changed passages of the manuscript are shown in cursive type. All pages and lines are in reference to the unmarked manuscript version: titled “Manuscript”.

Reviewer #1

Dear authors,

thank you very much for the opportunity to review the submitted manuscript. The authors conducted a secondary analysis of 108 critical care patients and analyzed the cardiorespiratory responses to rehabilitation exercises.

I am appreciating this work and have got some minor concerns.

We thank the reviewer for taking the time to review our manuscript and are grateful for the constructive and valuable feedback. All comments have been answered below.

SERIOUS CONCERNS

MINOR CONCERNS

Page 2, Line 30-33: wording: you used the terms “early rehabilitation”, “exercise”, “rehabilitation”, “training” and “recovery” as synonyms, I guess to improve readability. With a closer look, all those terms are different concepts, requiring different scientific approaches. Similar, I found in section introduction (p4, L105-110), that you used the term “early physiotherapy-led rehabilitation” here, but also “early mobilisation”, “exercise”, and “sessions”. At all, I suggest using one (1) term stringently.

We agree with the reviewer that these terms include different concepts. To improve clarity, we reworded the abstract and introduction section using the term ‘rehabilitation’ as suggested. We took this even further and reworded, where appropriate, throughout the manuscript.

We did, however, keep the term ‘recovery’ because it also includes the period after rehabilitation which was an important aspect of our analysis.

The sections mentioned by the reviewer now read:

Abstract (page 2, lines 30-33):

“Early rehabilitation is indicated in critically ill adults to counter functional complications. However, the physiological response to rehabilitation is poorly understood. This study aimed to determine the cardiorespiratory response to rehabilitation and to investigate the effect of explanatory variables on physiological changes during rehabilitation and recovery.”

Introduction (page 4, lines 104-107):

“First, we aimed to analyse physiological variables from before, during and after rehabilitation to describe both the response to rehabilitation and to estimate recovery. Second, we aimed to characterise effects of a-priori selected explanatory variables on cardiorespiratory reactions from before to during rehabilitation and from before to after rehabilitation.”

P2, L37: would you please add the information of the seven different session types, with increasing activities. Please use a stringent reporting.

We are unfortunately limited by the allowed word count within the abstract (maximal 300 words, initially 292 words). We therefore cannot add all explanatory variables. Moreover, to add only ‘session type’ would unduly elevate the variable above the others. Our hypotheses about the main drivers of the cardiorespiratory response to early rehabilitation also included mobilisation level, exercise modality and duration. These explanatory variables should therefore be considered as important as session type. It is also important to note, that the seven session types are not reflecting increasing activities. They are based on commonly used, combined interventions in clinical practice. For example, early mobilisation is often combined with respiratory management, because sitting on the edge of bed might help to clear secretions or because deep-breathing exercises might calm patients after an effort to sit up. More detailed information on the seven session types is provided within the supplement (S1 File).

However, we concede that the term rehabilitation is used differently across the world and that some clarity of our concept might be needed. While our manuscript includes a definition of rehabilitation (page 17, lines 348-349), we agree that the concept must be clear within the abstract. We therefore slightly changed the suggested sentence to include the pragmatic study design and some type of interventions within the abstract (page 2, lines 37-38):

“…, we analysed the 716 physiotherapy-led, pragmatic rehabilitation sessions (including exercise, cycling and mobilisation).”

P2, Abstract: I am fan of clear structures, e.g. using always the same order of reported outcome parameters: MV, VO2, MAP. Within the sentences, this order changes. Would you please re-arrange the order, and keep it? See also lines 153-156.

We have amended the order throughout the manuscript to ‘HR, MAP, MV, SpO2 and VO2’ when it made sense to do so. For example, we did not change the order, when the magnitude of the change was discussed (e.g., page 10, lines 246-247), but changed tables 2 and 3 to the same order. When one variable was not discussed we left this variable out, for example, we wrote ‘HR, MV and VO2’.

P2, Abstract: I guess all results are mean differences? If so, please add this information.

Indeed these are mean differences, we added this description to the results in the abstract (page 2, lines 49-51):

“Active patient participation increased MV (mean difference 0.7l/min [0.4-1.0, p<0.001]) and VO2 (23ml/min [95%CI: 13-34, p<0.001]) during training when compared to passive participation.”

P3, keywords: to improve your future citation, please check for pubmed’s MeSH terms (https://www.ncbi.nlm.nih.gov/mesh) for the used keywords, and schedule key words in alphabetically order.

We re-ordered keywords alphabetically. Five out of the nine keywords are MeSH-terms:

1. Critical Illness [C23.550.291.625]

2. Early ambulation (E02.760.169.063.500.335, E02.831.335)

3. Exercise [I03.350]

4. Oxygen Consumption [G03.680]

5. Physical Therapy Specialty [H02.010.625]

We further added the MeSH-term: Rehabilitation [E02.831], because this term contains the concept of recovery. We kept the terms physiotherapy; physiological reaction; recovery and training response because they are commonly used synonyms and thus complement the MeSH-terms. The manuscript has been changed as follows:

Page 3, lines 71-72:

“critical illness; early ambulation; exercise, oxygen consumption; physical therapy specialty; physiological reaction; physiotherapy; recovery; rehabilitation; training response”

L123: this was an RCT, but you analyse intervention- and control group as one single group? If so, please add some information about the intervention and an explanation why the summary is reasonable (if I remember right, the intervention group had a little bit more minutes of rehab each day, right?)

Yes, the reviewer’s assumption is correct. Our reasoning is as follows: This is a secondary analysis of an RCT that did not show any significant or clinically relevant differences in the prespecified primary and secondary outcomes. The two groups’ intervention did therefore not affect outcomes. In consequence the two treatments can be considered equal, which allowed us to combine both groups into one population to answer our research questions. The current dataset therefore includes the whole original population using data from all trial participants who had at least one rehabilitation session. 

We slightly revised the manuscript to clarify this part (page 5, lines 121-125)

“No significant differences were found in the primary or secondary outcomes with the exception of improved mental health six months after hospital discharge for the experimental group [19]. Consequently, this secondary analysis considered the two randomised groups as one population using data from all trial participants with at least one rehabilitation session.

Nevertheless, we did not ignore the allocated intervention within our analysis. Again, the reviewer is correct that the intervention group had significantly longer rehabilitation sessions. Also, only the intervention group cycled. However, these factors are accounted for within our analysis by the explanatory variables ‘session duration’ and ‘session type’. We decided a-priori which explanatory variables to include within our model. These variables were based on clinical reasoning and earlier research to limit confounding (also see revised lines 196-197 based on comments of reviewer #2). During this process we considered a variable ‘randomisation group’. However, we discarded it to avoid multicollinearity within the model and because we were rather interested on how the variables themselves affected the cardiorespiratory response (e.g. ‘session duration’).

Nevertheless, based on the reviewer’s comments, we performed a sensititvity analysis that includes the variable ‘randomisation group’ in our model. Compared to our main analysis, the estimates barely changed for MAP, MV and VO2. While we found a significant effect for ‘randomisation group’ for HR (‘during’: mean difference 1.47bpm (0.17 to 2.78), p=0.029; ‘after’: 2.1bpm (0.79 to 3.41), p=0.002), the estimates for our chosen explanatory variables changed only slightly when compared to our main analyses. Most importantly, our results and conclusions do not change but rather reinforce our earlier reasoning to exclude this variable. We therefore limit the results of this sensitivity analysis to this letter, but did not add them to the supplementary material. 

Outcome Explanatory variable Mean differences Lower 95% CI Upper 95% CI p-value

HR during

 Age (years) 0.02 -0.02 0.06 0.329

 Gender (male is reference) 0.21 -0.9 1.32 0.706

 Body Mass Index (kg/m2) -0.01 -0.11 0.1 0.908

 Randomisation group: intervention 1.47 0.17 2.78 0.029

 Daily SOFA score (0-24) 0 -0.12 0.12 0.959

 Session duration (min) 0.05 0.01 0.09 0.009

 Time from ICU admission to start of session (days) 0.03 -0.01 0.07 0.179

 Session type: cycling -1.66 -2.99 -0.33 0.015

 Session type: mobilisation 2.04 -1.88 5.95 0.309

 Session type: respiratory management 0.58 -1.11 2.27 0.501

 Session type: exercise and respiratory management 0.39 -1.06 1.84 0.597

 Session type: complex cycling and mobilisation -3.22 -6.45 0.02 0.052

 Session type: complex exercise and mobilisation 2.35 -1.45 6.15 0.226

 Treatment modality: mixed -0.45 -2.45 1.56 0.661

 Treatment modality: active 1.18 0.13 2.23 0.028

 Mobilisation level: edge-of-bed -0.98 -4.85 2.89 0.621

 Mobilisation level: out-of-bed 0.08 -3.98 4.14 0.97

 Airway support: tracheostomy -1.02 -2.52 0.49 0.186

 Airway support: tube -0.53 -1.84 0.79 0.433

 Opiates on session day 0.27 -1.21 1.75 0.719

 Vasoactive on session day -0.2 -1.15 0.74 0.676

 Sedatives on session day 1.28 0.19 2.37 0.021

 Neuromuscular blocking agents on session day -0.21 -1.3 0.88 0.705

HR after

 Age (years) 0.05 0.01 0.09 0.012

 Gender (male is reference) 0.49 -0.54 1.52 0.36

 Body Mass Index (kg/m2) -0.09 -0.19 0.01 0.084

 Randomisation group: intervention 2.1 0.79 3.41 0.002

 Daily SOFA score (0-24) -0.13 -0.26 -0.01 0.035

 Session duration (min) 0.01 -0.04 0.06 0.686

 Time from ICU admission to start of session (days) 0.04 0 0.08 0.046

 Session type: cycling -1.88 -3.42 -0.34 0.017

 Session type: mobilisation 6.66 2.04 11.27 0.005

 Session type: respiratory management 0.27 -1.68 2.22 0.785

 Session type: exercise and respiratory management 0.28 -1.41 1.97 0.745

 Session type: complex cycling and mobilisation -3.99 -7.79 -0.19 0.04

 Session type: complex exercise and mobilisation 4.82 0.34 9.31 0.035

 Treatment modality: mixed -1.03 -3.38 1.32 0.391

 Treatment modality: active 1.02 -0.18 2.22 0.097

 Mobilisation level: edge-of-bed -6.34 -10.91 -1.77 0.007

 Mobilisation level: out-of-bed -6.21 -10.98 -1.45 0.011

 Airway support: tracheostomy -0.17 -1.85 1.51 0.843

 Airway support: tube 0.81 -0.71 2.33 0.299

 Opiates on session day 0.29 -1.41 1.99 0.737

 Vasoactive on session day 0.07 -1.01 1.15 0.899

 Sedatives on session day 1.63 0.37 2.89 0.012

 Neuromuscular blocking agents on session day 0.09 -1.17 1.36 0.886

MAP during

 Age (years) 0.01 -0.03 0.05 0.615

 Gender (male is reference) 1.02 0 2.04 0.051

 Body Mass Index (kg/m2) 0.03 -0.08 0.14 0.575

 Randomisation group: intervention 0.28 -1.07 1.64 0.683

 Daily SOFA score (0-24) -0.02 -0.15 0.11 0.737

 Session duration (min) -0.06 -0.12 -0.01 0.017

 Time from ICU admission to start of session (days) 0.03 -0.02 0.07 0.242

 Session type: cycling 0.26 -1.4 1.92 0.761

 Session type: mobilisation 0.29 -4.66 5.25 0.907

 Session type: respiratory management 1.78 -0.4 3.96 0.109

 Session type: exercise and respiratory management 0.27 -1.54 2.07 0.772

 Session type: complex cycling and mobilisation 1.8 -2.26 5.86 0.386

 Session type: complex exercise and mobilisation 1.31 -3.48 6.11 0.592

 Treatment modality: mixed 2.87 0.25 5.49 0.032

 Treatment modality: active 1.42 0.14 2.7 0.03

 Mobilisation level: edge-of-bed 0.8 -4.1 5.7 0.749

 Mobilisation level: out-of-bed 0.79 -4.38 5.96 0.764

 Airway support: tracheostomy 0.9 -0.94 2.74 0.338

 Airway support: tube 1.14 -0.53 2.8 0.181

 Opiates on session day 0.6 -1.42 2.62 0.561

 Vasoactive on session day -1.82 -2.98 -0.66 0.002

 Sedatives on session day 1.38 -0.02 2.77 0.054

 Neuromuscular blocking agents on session day -0.39 -1.75 0.96 0.568

MAP after

 Age (years) 0 -0.05 0.05 0.977

 Gender (male is reference) 0.5 -0.75 1.74 0.432

 Body Mass Index (kg/m2) -0.13 -0.26 0 0.054

 Randomisation group: intervention -0.01 -1.66 1.64 0.99

 Daily SOFA score (0-24) -0.2 -0.36 -0.04 0.013

 Session duration (min) -0.01 -0.07 0.05 0.735

 Time from ICU admission to start of session (days) 0.01 -0.05 0.06 0.727

 Session type: cycling -0.36 -2.38 1.67 0.728

 Session type: mobilisation -1.78 -7.81 4.26 0.564

 Session type: respiratory management -0.69 -3.34 1.96 0.61

 Session type: exercise and respiratory management 0.32 -1.87 2.52 0.773

 Session type: complex cycling and mobilisation 0.71 -4.23 5.65 0.778

 Session type: complex exercise and mobilisation -0.35 -6.19 5.49 0.907

 Treatment modality: mixed 0.18 -3.02 3.37 0.914

 Treatment modality: active 1.44 -0.13 3 0.072

 Mobilisation level: edge-of-bed -1.35 -7.31 4.62 0.658

 Mobilisation level: out-of-bed -2.62 -8.91 3.68 0.415

 Airway support: tracheostomy 0.51 -1.74 2.75 0.659

 Airway support: tube 1.16 -0.86 3.19 0.261

 Opiates on session day -0.06 -2.56 2.43 0.96

 Vasoactive on session day -2.42 -3.83 -1 0.001

 Sedatives on session day 1.02 -0.68 2.73 0.239

 Neuromuscular blocking agents on session day -0.08 -1.73 1.57 0.926

MV during

 Age (years) 0 -0.01 0.01 0.823

 Gender (male is reference) -0.43 -0.74 -0.12 0.008

 Body Mass Index (kg/m2) -0.01 -0.04 0.02 0.366

 Randomisation group: intervention 0.31 -0.08 0.7 0.12

 Daily SOFA score (0-24) 0.01 -0.02 0.05 0.429

 Session duration (min) -0.02 -0.03 0 0.011

 Time from ICU admission to start of session (days) 0 -0.01 0.02 0.745

 Session type: cycling -0.02 -0.42 0.37 0.912

 Session type: mobilisation 0.8 -0.38 1.97 0.184

 Session type: respiratory management 0.31 -0.45 1.07 0.419

 Session type: exercise and respiratory management 0.03 -0.43 0.49 0.892

 Session type: complex cycling and mobilisation 0.01 -0.93 0.95 0.979

 Session type: complex exercise and mobilisation 0.29 -0.84 1.41 0.618

 Treatment modality: mixed 0.51 -0.1 1.12 0.104

 Treatment modality: active 0.73 0.4 1.06 0

 Mobilisation level: edge-of-bed 0.68 -0.49 1.86 0.256

 Mobilisation level: out-of-bed 0.37 -0.86 1.59 0.556

 Airway support: tracheostomy 0.29 -0.36 0.95 0.378

 Airway support: tube 0.08 -0.54 0.7 0.802

 Opiates on session day 0.09 -0.48 0.66 0.753

 Vasoactive on session day -0.15 -0.44 0.13 0.3

 Sedatives on session day 0.09 -0.3 0.47 0.668

 Neuromuscular blocking agents on session day -0.22 -0.54 0.1 0.18

MV after

 Age (years) 0 -0.01 0.01 0.653

 Gender (male is reference) -0.3 -0.58 -0.01 0.043

 Body Mass Index (kg/m2) -0.02 -0.05 0.01 0.2

 Randomisation group: intervention 0.18 -0.2 0.55 0.355

 Daily SOFA score (0-24) -0.01 -0.04 0.03 0.6

 Session duration (min) 0 -0.02 0.01 0.988

 Time from ICU admission to start of session (days) 0 -0.02 0.01 0.78

 Session type: cycling -0.16 -0.58 0.27 0.475

 Session type: mobilisation 1.18 -0.11 2.47 0.073

 Session type: respiratory management -0.87 -1.73 -0.01 0.048

 Session type: exercise and respiratory management 0.1 -0.4 0.6 0.7

 Session type: complex cycling and mobilisation 0.06 -0.96 1.08 0.911

 Session type: complex exercise and mobilisation 0.22 -1.01 1.45 0.723

 Treatment modality: mixed -0.44 -1.1 0.23 0.199

 Treatment modality: active 0.2 -0.16 0.56 0.273

 Mobilisation level: edge-of-bed -0.44 -1.73 0.84 0.497

 Mobilisation level: out-of-bed -1.18 -2.51 0.15 0.083

 Airway support: tracheostomy 0.29 -0.43 1.01 0.427

 Airway support: tube 0.21 -0.48 0.9 0.554

 Opiates on session day 0.35 -0.27 0.97 0.27

 Vasoactive on session day -0.3 -0.61 0.01 0.061

 Sedatives on session day 0.28 -0.15 0.71 0.203

 Neuromuscular blocking agents on session day -0.02 -0.37 0.33 0.909

VO2 during

 Age (years) -0.54 -0.88 -0.2 0.002

 Gender (male is reference) -23.7 -32.62 -14.78 0

 Body Mass Index (kg/m2) 0.84 -0.1 1.78 0.081

 Randomisation group: intervention 7.33 -6.46 21.12 0.299

 Daily SOFA score (0-24) 0.25 -0.83 1.32 0.654

 Session duration (min) -0.59 -1.08 -0.1 0.02

 Time from ICU admission to start of session (days) -0.16 -0.57 0.25 0.451

 Session type: cycling -1.97 -16.99 13.05 0.798

 Session type: mobilisation 37.81 -16.1 91.72 0.17

 Session type: respiratory management 2.6 -64.05 69.24 0.939

 Session type: exercise and respiratory management 0.99 -14.03 16.01 0.897

 Session type: complex cycling and mobilisation -12.78 -43.55 17.99 0.416

 Session type: complex exercise and mobilisation 20.3 -28.18 68.79 0.412

 Treatment modality: mixed 0.32 -19.89 20.53 0.975

 Treatment modality: active 22.89 11.75 34.02 0

 Mobilisation level: edge-of-bed -3.19 -54.93 48.55 0.904

 Mobilisation level: out-of-bed 7.99 -47.98 63.97 0.78

 Airway support: tracheostomy 2.54 -24.57 29.64 0.854

 Airway support: tube 2.58 -22.98 28.14 0.843

 Opiates on session day -7.66 -25.72 10.4 0.407

 Vasoactive on session day -7.39 -16.76 1.99 0.124

 Sedatives on session day 5.83 -8.8 20.47 0.435

 Neuromuscular blocking agents on session day -7.15 -17.59 3.29 0.18

VO2 after

 Age (years) -0.3 -0.68 0.08 0.121

 Gender (male is reference) -18.26 -27.94 -8.59 0

 Body Mass Index (kg/m2) 0.52 -0.5 1.55 0.318

 Randomisation group: intervention -0.72 -15.78 14.33 0.925

 Daily SOFA score (0-24) -0.32 -1.5 0.86 0.599

 Session duration (min) -0.28 -0.86 0.3 0.342

 Time from ICU admission to start of session (days) 0.07 -0.38 0.53 0.753

 Session type: cycling -1.81 -18.55 14.94 0.833

 Session type: mobilisation -12.14 -73.29 49 0.697

 Session type: respiratory management -16.09 -90.67 58.48 0.673

 Session type: exercise and respiratory management -4.26 -21.21 12.7 0.623

 Session type: complex cycling and mobilisation -11.11 -45.52 23.31 0.528

 Session type: complex exercise and mobilisation -12.72 -67.06 41.62 0.647

 Treatment modality: mixed -6.46 -29.04 16.12 0.575

 Treatment modality: active 12 -0.48 24.48 0.06

 Mobilisation level: edge-of-bed 13.5 -44.58 71.58 0.649

 Mobilisation level: out-of-bed -40.96 -104.24 22.33 0.206

 Airway support: tracheostomy 4.77 -25.47 35.01 0.757

 Airway support: tube 6.16 -22.37 34.69 0.672

 Opiates on session day -8.27 -28.2 11.67 0.417

 Vasoactive on session day -11.72 -22.17 -1.28 0.029

 Sedatives on session day 10.77 -5.85 27.4 0.205

 Neuromuscular blocking agents on session day -10.08 -21.74 1.58 0.091

L125 “physiological data (n=35)” You don’t mean that you analyzed 35 data, but data of 35 patients, right?

This is correct. We revised this to (page 5, lines 125-127):

“A preliminary safety analysis of the physiological data of the first 35 subjects indicated a moderately increased workload with increased heart rate and oxygen consumption but stable oxygen saturation from before to during rehabilitation [20].”

L126: please insert a line break between “…physiotherapy [20].” and “The local…”

This has been implemented as suggested.

L130: please add “Population”, and a line break. I guess you mean “equal or older than 18 years”?

We added a new heading ‘Population’ (line 132). Accordingly, we deleted ‘population’ from the heading ‘study design’. We further revised the sentence as follows (page 5, lines 133-134):

“Participants were ≥ 18 years old, functionally independent before ICU admission and expected to remain ventilated for ≥ 72 hours.”

Section results is fine. The tables are hard to read, but I have no idea how to improve readability. The “examples for interpretation” are really helpful.

As described previously, we changed the order of the column headings to HR, MAP, MV and VO2 to give a clear structure throughout the manuscript. We did not perform any other changes as readability should increase with article formatting. 

L316-319: would you please summarize the main results in their plain meaning, eg. “complex exercise and mobilisation led to increased VO2”, or else

We concede that tables 2 and 3 are complex. We believe that the examples for interpretation and the highlighted relevant 95% confidence intervals will help readers to interpret results. To further increase understanding, we give a full summary here (based on 95% confidence intervals) along with a brief summary within the manuscript.

Full summary for ‘during rehabilitation’

• HR: increased by ‘session duration’ (per each additional minute of therapy), active participation (when compared to passive therapy), and by sedatives (when compared to none).

• MAP: increased by female gender (when compared to male), mixed or active participation (when compared to passive therapy), decreased by vasoactive drugs (when compared to none) and ‘session duration’ (per each additional minute of therapy)

• MV: increased by active participation (when compared to passive therapy), decreased by female gender (when compared to male) and by ‘session duration’ (per each additional minute of therapy)

• VO2: increased by active participation (when compared to passive therapy), decreased by age (per each additional year of age), female gender (when compared to male) and ‘session duration’ (per each additional minute of therapy)

Full summary for ‘after rehabilitation’

• HR: increased by age (per each additional year of age), session types ‘mobilisation’ and ‘complex exercise and mobilisation’ (when compared to ‘exercise) and by sedatives (when compared to none), decreased by mobilisation level ‘edge-of-bed’ and ‘out-of-bed’ (when compared to ‘in-bed) and by daily SOFA score (per each additional point)

• MAP: decreased by BMI (per each additional unit), daily SOFA score (per each additional point) and by vasoactive drugs (when compared to none)

• MV: decreased by female gender (when compared to male)

• VO2: increased by mobilisation level ‘edge-of-bed’ (when compared to ‘out-of-bed’), decreased by female gender (when compared to male) and by vasoactive drugs (when compared to none)

We newly summarised this in the manuscript as follows (page 16, lines 333-338):

“Specifically, clinicians should be aware that during rehabilitation cardiorespiratory parameters (HR, MAP, MV and VO2) increase when using active participation versus passive therapy and decrease (with the exception of HR) with each additional minute of rehabilitation. In contrast, HR remains elevated in the recovery phase after (multimodal) mobilisation when compared to exercise and drops following higher mobilisation levels when compared to in-bed rehabilitation.”

Additionally, a summary is available within the results section (page 11, lines 275-279):

“Shorter ‘session duration’ and ‘active treatment modality’ generally increased physiological parameters during rehabilitation, while cardiorespiratory parameters did not return back to baseline for ‘session type’ and ‘mobilisation level’ during the prespecified 15-min recovery-phase. Explanatory variables with an effect on SpO2 were ‘mobilisation level’ and ‘airway support’ (S2 File: S4 Table).”

Section discussion is fine and interesting.

L339, 346, 358, 369, 388: any new hypothesis or research questions?

Yes, there are several research questions arising from our analysis. Most importantly, we found that ‘treatment modality’ and ‘session duration’ were the major drivers of cardiorespiratory changes during early rehabilitation. However, a sufficient cardiorespiratory response does not automatically translate into an adequate neuromuscular response or functional benefits. Consequently, future trials should investigate if any of these factors can improve patient-centred, functional outcomes. For example, shorter sessions were associated with an increased cardiorespiratory response. We propose to investigate whether shorter, more frequent sessions improve functional outcomes, when compared to only one session per day. 

We revised the manuscript to include this more clearly (page 17, lines 363-365):

“This strategy was associated with improved 3-month outcomes after stroke [27] and should be investigated in future randomised controlled trials.”

Patient participation (treatment modality) is dependent upon sedation but also on patients’ motivation or their perceived rate of exertion. There are a few studies who specifically investigate the impact of sedation on patient participation and functional outcomes [1], but patient fatigue is often a limiting factor [2]. A qualitative study that explores enabling factors and barriers to being active while critically ill might give insights into how future clinical trials should be shaped to achieve intensity-targets.

Additionally, we suggest to perform prospective studies that explore the optimal target for cardiorespiratory parameters and how this relates with patients’ perceived rate of exertion as well as neuromuscular activation (proof-of-concept study). For example, we chose a conservative cut-off of 10% for our analysis. However, this might not be sufficient to elicit a neuromuscular benefit. We would therefore propose to investigate the feasibility of different physiological targets (e.g., 10% versus 20% versus 30%) preferably using oxygen consumption. (Rationale: oxygen consumption seems the most important variable, when talking about exercise that is defined as a planned, structured, repetitive bodily movement produced by skeletal muscles that results in energy expenditure and aims to improve or maintain one or more components of physical fitness [3]). Research questions would include:

• Can patients achieve and sustain these intensities? 

• Are higher intensities associated with improved muscle activation (for example when measured with electromyography)?

• If these physiological targets prove feasible (and remain safe as within our study), what is the effect of these different targets on functional outcomes?

We revised the manuscript to express some of these thoughts (page 19, lines 423-425):

“Future trials should therefore investigate the feasibility and efficacy of different physiological training intensities as well as their association with neuromuscular activation and patients’ perceived rate of exertion.”

Finally, our findings should be validated in a prospective study with prospectively planned hypotheses (also see revision based on comments of reviewer #2: lines 418-419).

Not part of the review and just a suggestion: You started very early, at day 2, and the muscle loss might not be severe in early days. I wonder if there are any differences in cardiorespiratory responses to PT’s rehab when delivered within early <7 days or late ≥7 days?

Our analysis accounted the time from ICU admission to each session (median of 9 days with an interquartile range from 4 to 20 days, which includes the 7 day difference). The estimated effect on cardiorespiratory parameters was very minimal, clinically irrelevant and not statistically significant [HR: 0.02 (-0.01, 0.06); MAP: 0.03 (-0.02, 0.07); MV: 0.001 (-0.01, 0.01), VO2: -0.18 (-0.59, 0.20)]. Accordingly, we concluded that an early or late session did not affect cardiorespiratory parameters (page 17, lines 341-343).

However, cardiorespiratory parameters are a poor surrogate measure for muscle mass, thus to infer about potential neuromuscular effects, a study using electromyography might be necessary.

Very fine work, thank you.

Reviewer #2

Thank you for the opportunity to review this paper. The authors present the cardiorespiratory response in critically ill adult patients in response to various early rehabilitation interventions. This paper is well written and contributes novel data to the field, highlighting that early rehabilitation is safe.

The authors adequately identify limitations to the study. The heterogeneity in both the population and interventions and the impact on the conclusions is mentioned.

We thank the reviewer for taking the time to review our manuscript and are grateful for the constructive and valuable feedback. All comments have been answered below.

I have detailed a few minor points below:

1. It may be valuable to clarify/better define the "explanatory" variables. How were these decided on or chosen?

We agree, that this is a highly relevant point. As mentioned within the comments of Reviewer #1, we extensively discussed parameters within the research team and weighted them against the previous evidence. We modified the manuscript accordingly (page 7, lines 196-197):

“Explanatory variables were prospectively determined using extensive clinical reasoning and previous evidence to account for confounders. They included…”

2. If understood correctly, the clinically relevant cutoff of >10% is related to safety; however further justification as to why this was chosen and not 20% is required. This will perhaps be important when weighing up safety versus adequate intensity to bring about physiological change. In the discussion it seems as if this clinically relevant response is linked to response to exercise and not safety – please clarify.

Indeed, we chose the 10% cut-off based on safety recommendations that generally consider an increase of >20% as an adverse event. To the best of our knowledge, there is currently no guidance on the dose and safety of any training intensity available in the literature. We therefore chose the more conservative cut-off that was considered safe in previous studies. This is clearly a limitation, that we acknowledge within the manuscript. 

As the reviewer mentions, 20% might be a more adequate intensity to elicit a neuromuscular response potentially improving functional outcomes. To this end, we performed a sensitivity analysis using a 20% cut-off (S2 File: S2 Table), whereby 1 out of 10 sessions was affected. Adverse events within our trial were not based on fixed cut-offs, but rather on individually set limits by the treating physician. Overall, we reported only few, transient adverse events. Accordingly, we believe that a 20%-threshold might be safe and should be considered in future trials as a potential training intensity. We further included this in the manuscript which has been revised to:

Page 19, lines 421-425:

“Finally, the cut-off for a clinically relevant variation was chosen in absence of previous evidence. However, a cut-off of 20% still led to variations in 1 out of 10 sessions, particularly for MV and VO2 (S2 File: S2 Table). Our interventions were safe with only few, transient adverse events. Future trials should therefore investigate the feasibility and efficacy of different physiological training intensities as well as their association with neuromuscular activation and patients’ perceived rate of exertion.”

Additionally, we slightly revised the following passages to clarify that this cut-off was primarily about safety:

Page 8, lines 214-215:

“Thus, a 10% threshold might be a safe cardiorespiratory training intensity in the critically ill adult.”

Page 17, lines 352-353:

“We found that a clinically relevant variation was achieved in 1 out of 4 sessions.”

Additionally, it is important to note that the main aim of this study was to describe and explore the physiological response (changes) by early rehabilitation. The response to exercise is therefore an important topic within our discussion. To avoid any confusion between terms, we further substituted ‘clinically relevant change’ to ‘clinically relevant variation’ throughout the manuscript.

Finally, we incorporated subheadings that align with the results section to clarify that the 10% cut-off belongs to the safety analysis. The slightly revised section about the cut-off definition has been moved from the last paragraph of the heading ‘Data collection and measurements’ to the heading ‘statistical analysis’ (page 8, lines 208-215):

“Safety analysis

We used a mixed-effects logistic regression model which accounted for correlation of within-individual measurements to investigate explanatory variables related to clinically relevant variations and report odds ratios with 95% confidence intervals (CI). A clinically relevant variation in physiological measurements was defined as >10% variation– based on half of what is commonly reported as an adverse event (>20%) in the literature [22]. Exercise is structured and repetitive physical activity that results in energy expenditure [23]. Thus, a 10% threshold might be a safe cardiorespiratory training intensity in the critically ill adult.”

3. Stronger motivation is needed for not adjusting for multiple testing is needed.

We concede that the chance of false-positive findings is increased without adjustment for multiple testing. However, this is a secondary, hypothesis-generating analysis and not a primary hypothesis-testing study (null versus alternative hypothesis). Adjustments for multiple testing might therefore discard potentially useful observations [4, 5]. We therefore do not think that multiple testing is appropriate for our type of analysis. However, we concede that we need to clearly state this limitation to readers.

We therefore revised the manuscript as follows:

Page 8, lines 216-217:

“Considering the exploratory, hypothesis-generating purpose of our secondary analysis, the significance threshold was set to 0.05 without adjustment for multiple testing.”

Page 19, lines 415-416: 

“Additionally, the estimated effect of explanatory variables should be interpreted as hypothesis-generating and in the context of safety.”

Page 19, lines 418-419:

“These results therefore provide important information for future trials, but need to be validated in prospective studies.”

4. In Table 1, please could you clarify what is meant by "Time from ICU admission to start of each session (days)"?

We investigated 716 rehabilitation sessions. The 108 participants had a median of 3 [2-8] sessions (line 226). Accordingly, the time to each session varied in respect to ICU admission. For example, the first session could have been on the first day after admission, the second on the second day and the third session on the fifth day. This variable accounts for these different time periods. 

We renamed this variable to “Time from ICU admission to start of individual session (days)” and added a footnote to better explain the variable (Page 10, lines 234-235):

“ a number of sessions varied between patients, this variable takes into account the time from ICU admission to the start of each, individual session in the individual patient.”

5. While recommendations are made for clinical practice in terms of monitoring and recovery, what may be more helpful to clinicians and advancing the cause of early rehabilitation in the ICU is know how these physiological responses link to various outcomes. As highlighted by the authors, there is a gap in terms of adequate prescription (FITT) in the critically ill patients. Such data sets, could help shed light on factors such as the frequency and intensity of rehabilitation interventions to effect change on outcomes.

Authors highlight these results enhance clinical decision making around frequency and intensity, however, I think without looking at outcomes – this enhancement is limited – it may be safe, but there is no way to know it is effective with the data provided.

We agree with the reviewer that studies linking physiological responses to functional outcomes are needed. However, before this research can be conducted safety should be established. Additionally, we would need proof-of-concept studies, for example, is a higher intensity associated with a better neuromuscular response? However, our data-set was primarily obtained to answer the question about the efficacy of two different interventions. This analysis would not be in line with our initial research questions and conclusions. We therefore advocate for future prospective studies. We integrated this within the manuscript (page 19, lines 423-425):

“Future trials should therefore investigate the feasibility and efficacy of different physiological training intensities as well as their association with neuromuscular activation and patients’ perceived rate of exertion.”

We further added this limitation to the manuscript (page 19, lines 419-420):

“Fourth, while our results inform clinical decision-making on the intensity and duration of early rehabilitation, they cannot establish the effect on functional outcomes.”

Reviewer #3

This manuscript is a straightforward secondary analysis of data generated from a randomized controlled trial, investigating the effect of explanatory variables on physiological changes during training and recovery. While the study (and the analysis) looks timely, relevant, and on target, I state my thoughts below, primarily on the statistical analysis presented.

We thank the reviewer for taking the time to review our manuscript and are grateful for the constructive and valuable feedback. All comments have been answered below.

1. A multi-level logistic regression was proposed. However, it was not clear from the writeup (Statistical Analysis section) on how exactly the dichotomization of the response variable was considered, and how the multi-level part was handled? Was it via a generalized estimating equations, or via some mixed-effects models? A clear writeup is expected.

We apologise for the imprecise formulation of our multiple analyses. To clarify, we introduced three sub-headings within the statistical analysis chapter (analogous to the sub-headings within the results section):

• Descriptive analysis (line 186)

• Cardiorespiratory response (line 191)

• Safety analysis (line 208)

Our primary analysis (cardiorespiratory response) used an ANCOVA approach with Gaussian mixed-effects models which accounted for the fact that the measurements within the same individual are correlated. We changed the statistical section to (page 7, lines 192-195):

“The impact of explanatory variables on physiological values during (training) and after (recovery) rehabilitation was investigated in respect to before the physiotherapy session and variability (CV) with Gaussian mixed-effects models which accounted for correlation of within-individual measurements (S1 File) based on Vickers et al. [21].”

Safety analyses were also based on a mixed-effects (we changed the wording multilevel to mixed-effects for consistency) regression model approach for a binary outcome. We corrected the statistical section to (page 8, lines 209-210):

“We used a mixed-effects logistic regression model which accounted for correlation of within-individual measurements to investigate…”

2. Tables 2 and 3 summarizes the results (estimated fixed effects of explanatory variables); it was not clear whether the table entries (estimates) are the log(oods-ratios), or something else? This needs to be made clear in the Table captions.

Again, we apologize for the unclear write-up of our statistical analysis section that we revised to include sub-headings.

Table 2 and 3 report the fixed effect estimates from Gaussian mixed-effect models (e.g. from the cardiorespiratory response). Accordingly, the estimates in Table 2 and 3 are mean differences of median values (i.e. median values of 2 min recorded values for each session). We changed the legend caption to clarify this (page 13, line 287 and page 15, line 301):

“Legend: Reported effects of explanatory variables are mean differences of median values and…”

The safety analys which used mixed-effect logistic regression models reports odds ratio (see S3 File and page 16, lines 317-320):

“For example, patients with an out-of-bed/edge-of-bed mobilisation had a 2.3 times higher chance (odds ratio 95%-CI 0.4-13.4, p=0.36) to demonstrate a clinically relevant 10% variations in ‘VO2 during’ and a 4.7 times higher chance (95%CI 0.46-48.42, p=0.19) in ‘VO2 after’ compared to an in-bed mobilisation (S3 File).”

3. How about assessing the goodness-of-fit (GOF) after these multi-level logistic regression fits? While the Hosmer-Lemeshow is ideal for assessing the standard logistic regression fit, there exists various proposals for the multi-level setup; see below:

The authors may consider producing some summary GOF statistics in this context.

https://digscholarship.unco.edu/cgi/viewcontent.cgi?article=1243&context=dissertations

Since our primary research question (cardiorespiratory response) was to investigate the association of a priori defined factors on study outcomes (and not predictions), we did not report GOF measures. In the table below, we provide the reviewer the GOF measures R-squared (R2), Akaike information criterion (AIC) and the root mean square error (RMSE). For all study outcomes we found R2 values ranging from 0.796 to 0.927. RMSE ranged from 1.11 to 35.9. 

Outcome GOF measure Value

HR during R2 0.927

 AIC 3376

 RMSE 4.03

HR after R2 0.904

 AIC 3541

 RMSE 5.02

MAP during R2 0.826

 AIC 3389

 RMSE 5.39

MAP after R2 0.76

 AIC 3582

 RMSE 6.56

MV during R2 0.858

 AIC 1528

 RMSE 1.11

MV after R2 0.836

 AIC 1583

 RMSE 1.28

VO2 during R2 0.871

 AIC 3013

 RMSE 31.4

VO2 after R2 0.796

 AIC 3035

 RMSE 35.9

The reporting of GOF measures likely increases the complexity of the manuscript further (see comment from Reviewer #1). In consequence, we decided not to include this table in the manuscript. 

4. Interpretation of covariates/explanatory variables in the Results section should be considered in terms of increase/decrease in odds, with associated 95% confidence intervals.

As highlighted in our comments above, Table 2 and 3 report mean differences of median values and not odds ratios as these results are based on our primary analysis (cardiorespiratory response). The results from the safety analysis are reported within S3 File using odds ratios as effect measures. We hope that we could clarify this by using different sub-headings within our statistical analysis chapter.

References

1. Schweickert WD, Pohlman MC, Pohlman AS, Nigos C, Pawlik AJ, Esbrook CL, et al. Early physical and occupational therapy in mechanically ventilated, critically ill patients: a randomised controlled trial. Lancet. 2009;373(9678):1874-82. Epub 2009/05/19. doi: 10.1016/S0140-6736(09)60658-9. PubMed PMID: 19446324.

2. Wright SE, Thomas K, Watson G, Baker C, Bryant A, Chadwick TJ, et al. Intensive versus standard physical rehabilitation therapy in the critically ill (EPICC): a multicentre, parallel-group, randomised controlled trial. Thorax. 2017. Epub 2017/08/07. doi: 10.1136/thoraxjnl-2016-209858. PubMed PMID: 28780504.

3. Caspersen CJ, Powell KE, Christenson GM. Physical activity, exercise, and physical fitness: definitions and distinctions for health-related research. Public health reports (Washington, DC : 1974). 1985;100(2):126-31. Epub 1985/03/01. PubMed PMID: 3920711; PubMed Central PMCID: PMCPMC1424733.

4. Streiner DL, Norman GR. Correction for multiple testing: is there a resolution? Chest. 2011;140(1):16-8. Epub 2011/07/07. doi: 10.1378/chest.11-0523. PubMed PMID: 21729890.

5. Althouse AD. Adjust for Multiple Comparisons? It's Not That Simple. Ann Thorac Surg. 2016;101(5):1644-5. Epub 2016/04/24. doi: 10.1016/j.athoracsur.2015.11.024. PubMed PMID: 27106412.

---

## [Decision Letter · Decision Letter 1]

16 Dec 2021

PONE-D-21-27687R1Cardiorespiratory response to early rehabilitation in critically ill adults: a secondary analysis of a randomised controlled trialPLOS ONE

Dear Dr. Eggmann,

Thank you for submitting your revised manuscript to PLOS ONE. We are generally happy with the revisions made in response to reviewer comments, however Reviewer 2 has identified a final minor revision, which is required in order to accept the manuscript for publication. Kindly make the required change, clarifying that "clinically relevant"  is related to safety  - as editor, I will personally review when you resubmit and if this correction has been done, will accept the manuscript.

We look forward to receiving your revised manuscript.

Kind regards,

Brenda M. Morrow, PhD

Academic Editor

PLOS ONE

Journal Requirements:

Reviewers' comments:

Reviewer's Responses to Questions

**Comments to the Author**

1. If the authors have adequately addressed your comments raised in a previous round of review and you feel that this manuscript is now acceptable for publication, you may indicate that here to bypass the “Comments to the Author” section, enter your conflict of interest statement in the “Confidential to Editor” section, and submit your "Accept" recommendation.

Reviewer #1: All comments have been addressed

Reviewer #2: All comments have been addressed

Reviewer #3: All comments have been addressed

2. Is the manuscript technically sound, and do the data support the conclusions?

Reviewer #1: Yes

Reviewer #2: Yes

Reviewer #3: (No Response)

3. Has the statistical analysis been performed appropriately and rigorously? 

Reviewer #1: Yes

Reviewer #2: Yes

Reviewer #3: (No Response)

4. Have the authors made all data underlying the findings in their manuscript fully available?

Reviewer #1: Yes

Reviewer #2: Yes

Reviewer #3: (No Response)

5. Is the manuscript presented in an intelligible fashion and written in standard English?

Reviewer #1: Yes

Reviewer #2: Yes

Reviewer #3: (No Response)

6. Review Comments to the Author

Reviewer #1: Dear authors

Thank you very much for the revised manuscript. You were able to address all concerns, or argued very reasonable. I have no further ideas how to improve the manuscript.

Again, very fine work, with a high relevance for practice!

Reviewer #2: Thank you for your comprehensive response and changes to the manuscript.

I think it would be good to clarify that "clinically relevant" (if I understand correctly) is related to safety - not sure this is entirely clear.

Perhaps line 431 should read "negatively effect physiological parameters" linking it back to safety. I think this is important as the next step would then be navigating the line between safety and efficacy (intensity)

Reviewer #3: (No Response)

7. PLOS authors have the option to publish the peer review history of their article (what does this mean?). If published, this will include your full peer review and any attached files.

Reviewer #1: No

Reviewer #2: No

Reviewer #3: No

---

## [Author Response · Author response to Decision Letter 1]

17 Dec 2021

Responses to Reviewers

The reviewers’ comments are shown in bold type, our responses in plain type and changed passages of the manuscript are shown in cursive type. All pages and lines are in reference to the revised manuscript in the provided PDF-proof.

Reviewer #2:

Thank you for your comprehensive response and changes to the manuscript.

I think it would be good to clarify that "clinically relevant" (if I understand correctly) is related to safety - not sure this is entirely clear.

Perhaps line 431 should read "negatively effect physiological parameters" linking it back to safety. I think this is important as the next step would then be navigating the line between safety and efficacy (intensity)

We appreciate the reviewer’s comment and changed the manuscript as follows:

Lines 211-213:

“The safety cut-off for a clinically relevant variation in physiological measurements was defined as >10% variation – based on half of what is commonly reported as an adverse event (>20%) in the literature [22].”

We would further like to mention, that above sentence is written under the subheading ‘safety analysis’ which should make it very clear to readers that this is about safety.

Lines 421-422:

“Finally, the safety cut-off for a clinically relevant variation was chosen in absence of previous evidence.”

Lines 431-432:

“[…], our findings indicate that rehabilitation in the ICU is safe and does not negatively influence physiological parameters.”

We would further like to highlight the following (unchanged) passage within our manuscript which elucidates that ‘clinically relevant variations’ are in relation to safety:

Lines 109-112:

“Finally, we re-evaluated safety by examining sessions with clinically relevant variations (>10%), an adverse event or therapy discontinuation to determine predictors and explored individual characteristics of patients with a strong physiological reaction.”

We thank the reviewer for their time and dedication to improve our manuscript. We hope to have fully satisfied all concerns in order to see this manuscript published.

---

## [Editor Report · Decision Letter 2]

5 Jan 2022

Cardiorespiratory response to early rehabilitation in critically ill adults: a secondary analysis of a randomised controlled trial

PONE-D-21-27687R2

Dear Dr. Eggmann,

We’re pleased to inform you that your manuscript has been judged scientifically suitable for publication and will be formally accepted for publication once it meets all outstanding technical requirements.

Kind regards,

Brenda M. Morrow, PhD

Academic Editor

PLOS ONE
---

## [Editor Report · Acceptance letter]

26 Jan 2022

PONE-D-21-27687R2 

Cardiorespiratory response to early rehabilitation in critically ill adults: a secondary analysis of a randomised controlled trial 

Dear Dr. Eggmann:

I'm pleased to inform you that your manuscript has been deemed suitable for publication in PLOS ONE. Congratulations! Your manuscript is now with our production department. 

Kind regards, 

on behalf of

Professor Brenda M. Morrow 

Academic Editor

PLOS ONE